# Learning When to Plan: Efficiently Allocating Test-Time Compute for LLM Agents

To our knowledge, this work is the first to explore fine-tuning LLM agents to internalize dynamic test-time compute allocation via RL in sequential decision-making tasks

## Abstract

Training large language models (LLMs) to reason via reinforcement learning (RL) significantly improves their problem-solving capabilities. In agentic settings, existing methods like ReAct prompt LLMs to explicitly plan before every action; however, we demonstrate that always planning is computationally expensive and degrades performance on long-horizon tasks, while never planning further limits performance. To address this, we introduce a conceptual framework formalizing dynamic planning for LLM agents, enabling them to flexibly decide when to allocate test-time compute for planning. We propose a simple two-stage training pipeline: (1) supervised fine-tuning on diverse synthetic data to prime models for dynamic planning, and (2) RL to refine this capability in long-horizon environments. Experiments on the Crafter environment show that dynamic planning agents trained with this approach are more sample-efficient and consistently achieve more complex objectives. Additionally, we demonstrate that these agents can be effectively steered by human-written plans, surpassing their independent capabilities. To our knowledge, this work is the first to explore finetuning LLM agents to internalize dynamic test-time compute allocation via RL in sequential decision-making tasks, paving the way for more efficient, adaptive, and controllable agentic systems.

## 1 Introduction

A key insight from recent work on LLM reasoning is the role of *test-time compute* — the ability to allocate additional computational resources to more difficult problems (Snell et al., 2025; Guo et al., 2025). For humans, difficult tasks often require deliberate thinking. Similarly, LLMs benefit from dedicating extra processing to explicitly reason through steps via chain-of-thought (Wei et al., 2022). In settings like math problem-solving and code generation, reasoning can enable models to explore possible answers before settling on a response in a manner akin to search (Xiang et al., 2025; Prystawski et al., 2023; Ruis et al., 2025). Reasoning LLMs trained to effectively use additional test-time compute on single-step tasks have also been shown to make extremely effective zero-shot agents (Yao et al., 2023b). However, a critical open question remains: *Can we further improve an LLM's ability to effectively allocate test-time compute on sequential decision-making tasks?* On the challenging agentic benchmark BALROG (Paglieri et al., 2025a), reasoning models have thus far only shown marginal gains over models immediately producing the next action (Paglieri et al., 2025b).

In agentic tasks, planning naturally emerges as a multi-step analogue to single-step chain-of-thought reasoning. Rather than committing immediately to a single next action, an agent can invest computational resources to better understand the current state and anticipate the outcome of future actions sequences. Crucially, unlike static single-turn tasks like MATH or GSM8K (Hendrycks et al., 2021; Cobbe et al., 2021), these agentic environments require maintaining plans over long horizons of interaction, executing multiple steps before replanning is required. Plans can serve as a guide for subsequent actions and improve the strategic coherence of future behaviour. However, introducing explicit planning presents its own critical challenge: deciding precisely when an agent should plan. This decision must carefully balance the performance improvements gained from more informed decision-making against the computational cost and additional variance in behaviour incurred by frequent replanning.

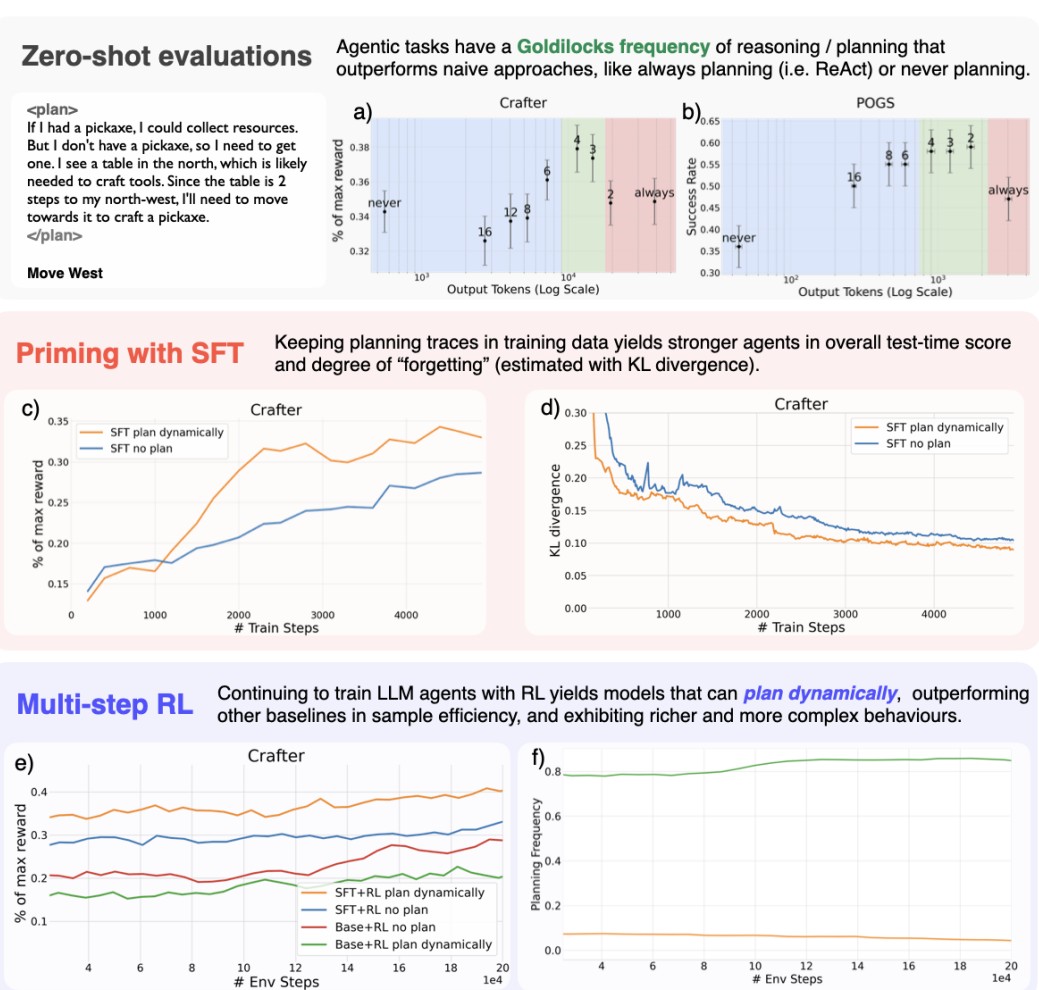

Figure 1: **Dynamic planning strategies across environments and training stages.** (a-b) Zero-shot results showing optimal "Goldilocks" planning frequency in Crafter and POGS (100 seeds, bars=standard-error). (c-d) SFT results demonstrating planning agents' improved performance with lower KL divergence from base model. (e-f) RL results where SFT-primed planning agents are more sample efficient than non-planning baselines and more consistently reach complex achievements.

To formalise this problem, we develop a framework for modelling the cost-benefit trade-offs of planning in partially-observable environments. According to our framework, agents should allocate test-time compute for planning only when anticipated improvements in policy performance outweigh the associated computational costs and any instability or noise induced by excessive replanning.

We experimentally investigate these concepts in two distinct environments: Partially-Observable Graph Search (POGS), a synthetic environment that we design to systematically evaluate planning abilities, and Crafter, a Minecraft-inspired grid-world environment (Hafner, 2022). Inspired by recent work showing that the presence of key inductive biases in training data are necessary for effective self-improvement (Gandhi et al., 2025), we develop a two-stage approach: first priming models with diverse planning behaviours through supervised fine-tuning (SFT), then applying RL. Using this approach, we successfully train agents that learn to plan strategically, execute their plans, and replan only when necessary, outperforming non-planning baselines trained via an equivalent two-stage pipeline. Furthermore, following the RL stage, agents that are trained to produce and follow their own plans can be effectively steered by plans produced by humans to achieve performance that the agents cannot reach alone. In summary, our experiments yield four key insights:

1. Each task has a "Goldilocks" frequency for planning that clearly outperforms naive strategies of always planning or never planning.

2. SFT priming demonstrates that including explicit natural language plans in training data significantly improves imitation learning compared to using identical action seqTAuences without plans.

3. RL fine-tuning after SFT priming yields planning agents that further outperform baselines in sample efficiency, and learn to plan, execute plans, and replan when necessary.

4. Planning agents can be collaboratively steered by humans that produce plans for them. This is only the case following RL and is not achieved by SFT priming alone.

Our work provides clear evidence that dynamic planning facilitates effective allocation of test-time compute in sequential decision making environments, showing that LLM agents can be trained to use additional computational resources intelligently. The ability to steer such planning agents—now capable enough to complete Crafter by collecting diamonds under human guidance—marks a significant step towards safer and more collaborative LLM agents. Together, these findings suggest a promising path towards more capable, efficient, interpretable, and steerable agentic systems.

## 2 RELATED WORK

**Classical Planning Methods**    Historically, much progress in sequential decision making has involved systems that explicitly look ahead before acting. Monte Carlo Tree Search (MCTS)(Coulom, 2006), combined with deep neural networks, has driven landmark systems like AlphaGo and MuZero(Silver et al., 2017b;a; Schrittwieser et al., 2020). Model Predictive Control (MPC) iteratively plans short horizons, adapting to new observations (Mayne et al., 2000). Similarly, model-based RL methods, such as World Models, PlaNet, and Dreamer, use imagination rollouts in latent space for effective planning and learning (Ha & Schmidhuber, 2018; Hafner et al., 2019; 2020). Collectively, these approaches underscore the strength of explicit planning, particularly when accurate internal or environmental models are available.

**LLM Reasoning and Planning**    Large language models have demonstrated significant reasoning capabilities, particularly through techniques like chain-of-thought (CoT) prompting (Wei et al., 2022). ReAct extends CoT into sequential settings, explicitly prompting models to reason before acting (Yao et al., 2023b), effectively enforcing an "always-plan" strategy. While methods like Tree of Thoughts (ToT) (Yao et al., 2023a) focus on optimizing the search strategy at a single decision point, dynamic planning addresses the orthogonal challenge of temporally allocating planning compute over a long-horizon trajectory. Similar reasoning methods include self-reflective prompting (Shinn et al., 2023; Wang et al., 2023; Hao et al., 2023; Besta et al., 2024; Yao et al., 2023a), automated prompt tuning (Fernando et al., 2024; Hu et al., 2025), and strategic planning demonstrated by CICERO in the Diplomacy game (, FAIR). However, frequent replanning can cause behavioural instability, analogous to RL frame-skipping strategies that advocate less frequent action repetition for improved exploration and consistency (Sharma et al., 2017; Kalyanakrishnan et al., 2021). Recent studies also show diminishing returns and increased brittleness from excessive reasoning (Stechly et al., 2024; Mizrahi et al., 2024; Sui et al., 2025), emphasizing the need for adaptive planning mechanisms. While concurrent works like Hu et al. (2024) and Qin et al. (2025) explore adaptive strategies, they typically rely on separate mediator policies to trigger external tools or SFT to imitate dynamic behaviours. In contrast, our approach optimizes a single monolithic LLM via RL to implicitly learn the cost-benefit trade-off of allocating its own test-time compute. Behaviour cloning with LLMs on data that includes textual reasoning between actions has also been shown to help imitation learning (Yang et al., 2022; Hu & Clune, 2023).

**Test-Time Compute Scaling**    More recently, test–time scaling has shown great promise, spearheaded by the results of OpenAI o1 (Jaech et al., 2024) and DeepSeek R1 (Guo et al., 2025). These gains arise when LLMs improve their own reasoning traces through RL training on tasks with verifiable rewards (Lambert et al., 2024). Methods such as STaR (Zelikman et al., 2022), Quiet-STaR (Zelikman et al., 2024), ScoRE (Kumar et al., 2025), and Meta-CoT (Xiang et al., 2025) showcase iterative self-improvement. Simple prompting strategies like s1 (Muennighoff et al., 2025) and critical insights from (Gandhi et al., 2025), demonstrating the necessity of supervised fine-tuning (SFT) priming with reasoning examples, further support this direction. Moreover, emergent planning capabilities have been observed from RL-trained base models as comments in code tasks (Zhao et al.,

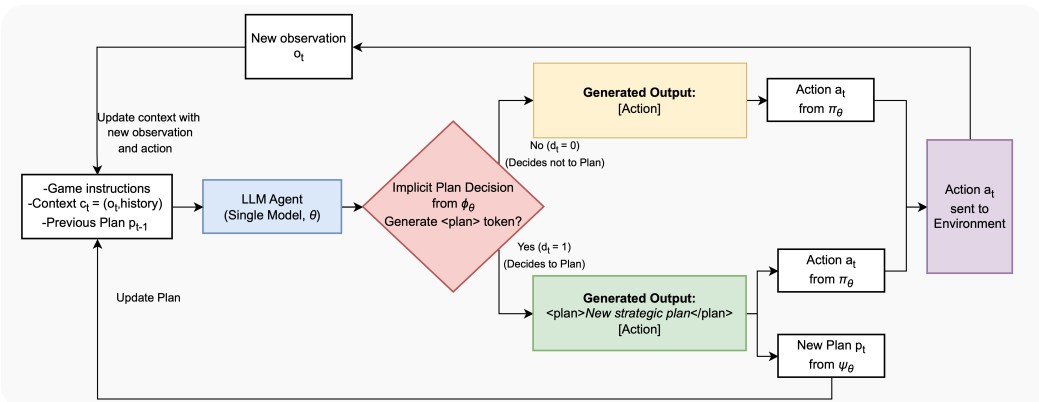

Figure 2: **Dynamic Planning Agent Architecture.** Our agent is a single, monolithic LLM whose conceptual policies are realized through its unified output format. The decision to plan ($\phi_\theta$) is made implicitly by the model's choice to begin its generation with a `<plan>` token. This single output string is then parsed to extract the action ($a_t$) and, if present, the new plan ($p_t$), thereby executing the acting ($\pi_\theta$) and planning ($\psi_\theta$) policies.

2025). While fixed always-planning hierarchical strategies exist (Erdogan et al., 2025), their rigidity motivates research toward adaptive, dynamic approaches.

**Steering LLM Agents** Recent studies have explored methods to steer LLM agents, such as influencing exploration through modulated representational uncertainty (Rahn et al., 2024), adaptively selecting reasoning modes based on task demands (Chen et al., 2024), and improving collaborative decision-making via step-wise RL evaluations (Zhou et al., 2025). Our work demonstrates that LLM agents can be effectively steered through adaptive planning, enabling integration of external human-generated plans post-RL training.

## 3 A CONCEPTUAL FRAMEWORK FOR DYNAMIC PLANNING WITH LLM AGENTS

Deciding when to allocate test-time compute for planning is a central challenge for LLM agents. To address this in a principled way, we first establish a conceptual framework that formalizes the underlying cost-benefit trade-offs. This framework provides the theoretical motivation for our practical training methodology, which uses reinforcement learning to teach an agent to implicitly master this dynamic planning skill.

Consider a sequential decision-making environment modelled as a Partially-Observable Markov Decision Process $\langle S, A, T, R, O, \gamma \rangle$ (states, actions, stochastic transitions, rewards, observations, discount factor). An LLM agent with parameters $\theta$ acts within this framework by generating tokens.

Specifically, at each timestep $t$, the agent receives an observation $o_t$, described in natural language, and maintains an internal context $c_t = (o_t, history)$ which includes the current observation, a history of previous observations and actions, and any existing plan $p_{t-1}$. Formally, the agent's behaviour is decomposed into a decision policy $\phi_\theta$, a planning policy $\psi_\theta$, and an acting policy $\pi_\theta$:

$$\phi_\theta(d_t \mid c_t, p_{t-1}), \quad \psi_\theta(p_t \mid c_t, p_{t-1}), \quad \pi_\theta(a_t \mid c_t, p_t)$$

Importantly, these three policies are not separate architectural components but rather a conceptual decomposition of the unified output from a single, monolithic LLM, no additional gating policies are added (Figure 2). The decision policy $\phi_\theta$ corresponds to the decision $d_t \in \{0, 1\}$, where $d_t = 1$ signifies that a new plan $p_t$ will be generated by the planning policy $\psi_\theta$. If $d_t = 0$, the agent continues with the existing plan $p_{t-1}$. Thus the plan selection mechanism is:

$$p_t = d_t \cdot \psi_\theta(p_t \mid c_t, p_{t-1}) + (1 - d_t) \cdot p_{t-1}$$

Finally, the acting policy $\pi_\theta$ generates action $a_t$ based on $c_t$ and $p_t$.

### 3.1 WHEN SHOULD AN AGENT PLAN?

Intuitively, an agent should only plan when the expected benefit outweighs its cost. We quantify this state-dependent trade-off using a simple cost-benefit analysis:

The expected benefit of planning, or the *Planning Advantage*, measures how much the agent's expected future rewards improve by adopting a new plan generated by $\psi_\theta$ (i.e., if the decision $d_t = 1$ is made), compared to continuing with the existing plan $p_{t-1}$. Conceptually, the value of generating a new plan is rooted in its potential to reduce the agent's uncertainty about optimal future actions and to augment the context. By making strategic reasoning explicit, a new plan provides actionable insights that go beyond what is implicitly encoded in the agent's internal representation (weights and activations). We formally define the planning advantage as the expected improvement in task-specific value, conditioned on the decision to generate a new plan:

$$A_{plan}(c_t) = \mathbb{E}_{p_t \sim \psi_\theta(\cdot | c_t, d_t = 1)}[V^{\pi_\theta}(c_t, p_t) - V^{\pi_\theta}(c_t, p_{t-1})]$$

Here, $V^{\pi_\theta}(c_t, p_t)$ represents the expected future rewards under the new plan $p_t$, and similarly for the existing plan $p_{t-1}$. While the agent does not explicitly compute $A_{plan}(c_t)$ at each step, its decision policy $\phi_\theta$ is trained to generate outputs that approximate this benefit, as detailed in Section 3.3.

The overall cost of planning, $C_{plan}$, arises from several sources:

$$C_{plan} = C_{tokens} + C_{latency} + C_{noise}$$

These components include:

**Computational Cost:** The direct cost of generating a plan, proportional to its token length: $C_{tokens} = k_{tokens} \cdot |p_t|$. This is a direct and measurable cost that we can explicitly penalize during training.

**Latency Cost:** The cost associated with the real-world time $\Delta T_{plan}$ taken to plan. This is included for theoretical completeness, as it is a critical factor in time-sensitive applications like robotics, where its impact would be implicitly absorbed by the task reward. However, in the turn-based environments used in our experiments (POGS and Crafter), the environment pauses for the agent's turn, so this cost is effectively zero ($C_{latency} \approx 0$).

**Instability Cost:** This is a conceptual cost representing the performance degradation that can arise from erratic or excessive replanning. Frequent replanning, especially with imperfect or inconsistent plans, can introduce behavioral instability (e.g., inefficient backtracking, subgoal oscillation) that ultimately hinders task success. We model this conceptually as $C_{noise} = k_{noise} \cdot f_p \cdot (1 - \bar{Q}_p)$, where the negative impact of high planning frequency ($f_p$) is magnified by low-quality plans (a low average plan quality $\bar{Q}_p$). This cost is not explicitly calculated during training; instead, its effects are implicitly penalized by the RL objective because they naturally lead to lower task rewards. Our backtracking analysis in POGS (Appendix B) serves as an empirical proxy for this instability.

### 3.2 PLAN DRIFT

The usefulness of an existing plan is not static; it typically diminishes over time as the agent acts and the environment evolves. This decay in relevance, or *plan drift*, makes replanning increasingly advantageous. Several factors contribute to how quickly plan drift occurs:

**Plan Abstraction Level:** High-level, conceptual plans (e.g., control the centre in chess) offer robustness against minor environmental shifts and remain relevant longer, though they provide less explicit guidance. Conversely, low-level detailed plans (e.g., specific move sequences) provide clearer direction but become outdated quickly.

**Planner and Model Accuracy:** Plans from highly accurate models tend to be robust and endure longer. In contrast, plans from imperfect models, like LLM natural language reasoning, may contain inaccuracies that accelerate their decay.

**Environment Dynamics:** The environment's volatility significantly influences plan lifespan. In stable environments, plans retain value longer, while in dynamic environments with unpredictable shifts (e.g., an opponent's unexpected move), existing plans can become instantly obsolete.

Understanding plan drift helps explain why agents must periodically reassess when to allocate compute to planning rather than following fixed planning strategies.

### 3.3 TRAINING THE DYNAMIC PLANNING AGENT

To enable the agent to learn when to plan, thereby implicitly performing the cost-benefit analysis outlined in Section 3.1, we use RL fine-tuning. The agent's policy parameters $\theta$ (governing the decision policy $\phi_\theta$, planning policy $\psi_\theta$ and acting policy $\pi_\theta$) are optimized to maximize the expected discounted sum of task rewards, adjusted by a penalty for the computational cost of planning:

$$\theta^* = \arg\max_\theta \mathbb{E}_{\tau \sim \theta} \left[ \sum_{t=0}^{H} \gamma^t \left( R_{task}(s_t, a_t) - d_t \cdot C_{tokens,t} \right) \right]$$

Other detrimental effects of poor planning strategies, such as those arising from excessive latency or instability (conceptualized as $C_{latency}$ and $C_{noise}$), are implicitly discouraged as they naturally lead to lower task rewards $R_{task}(s_t, a_t)$. Thus, by optimizing this objective, the decision policy $\phi_\theta$, planning policy $\psi_\theta$, and acting policy $\pi_\theta$ should jointly learn to optimally decide *when* to plan ($d_t = 1$), *how* to output plans ($p_t$) that are beneficial, and how to output actions ($a_t$) that effectively *follow* these plans, ensuring that the expected improvement in future task rewards (i.e., the empirical benefit corresponding to the conceptual $A_{plan}(c_t)$) outweighs the explicit cost $C_{tokens,t}$ as well as any implicit degradation of $R_{task}$ due to poor planning.

## 4 EXPERIMENTAL SETUP

Our experiments evaluate planning agents across diverse settings. In this section, we detail the environments used, the core evaluation protocol, and the specific setups for evaluation, SFT, and RL.

### 4.1 ENVIRONMENTS

To evaluate dynamic planning across different conditions, we select two complementary environments. First, **Partially Observable Graph Search (POGS)** is our custom synthetic environment designed to isolate planning under uncertainty. Agents navigate procedurally generated graph using only local observations, which require adaptive replanning upon discovery of new nodes or dead ends. POGS allows measurement of exploration efficiency via backtracking statistics. Second, **Crafter** (Hafner, 2022) is a complex 2D grid-world, long-horizon benchmark inspired by Minecraft. It demands multi-scale planning for survival, resource management, and crafting, testing both short-term tactical decisions and long-term strategic choices. Interaction in both environments occurs via natural language. Full technical details and figures for the environments are provided in the Appendix B.

### 4.2 EVALUATION PROTOCOL

We utilize the BALROG benchmark (Paglieri et al., 2025a) for standardized agent evaluation and environment interaction. At each timestep $t$, the agent receives its history and current observation $o_t$ within a chat template, guided by a system prompt outlining the task (Fig. 4 in Appendix A). The agent's response must include a natural language action command $a_t$. Our dynamic planning agents are instructed to decide at each step whether to plan. If they choose not to plan, they output only the action command `[Action]`. If they choose to plan, they output the plan followed by the action, using the format `<plan> [natural language plan] </plan> [Action]`. BALROG parses this output, identifying a planning decision ($d_t = 1$) if the `<plan>` block is present and using its content as the current plan $p_t$ in subsequent context. The `[Action]` command is always extracted and executed. Fallback mechanisms ensure robustness against invalid outputs. Appendix A provides detailed prompts.

### 4.3 ZERO-SHOT EVALUATION

To understand baseline capabilities and the raw effect of planning frequency, we perform zero-shot evaluations using Llama-3.3-70B-Instruct (Grattafiori et al., 2024) on POGS and Crafter (100 seeds

each). We compare different prompting strategies without any fine-tuning. We test a **Naive Agent**, prompted to only output actions and thus never plan. We also test **Fixed-Frequency Planners**, which are prompted to `plan-every-`$k$`-steps` for various $k \in \{1, 2, 4, 8, ...\}$; these agents are instructed to output a plan followed by an action every $k$ steps, and only an action otherwise. These evaluations measure performance trade-offs associated purely with inference-time planning strategies.

### 4.4 SUPERVISED FINE-TUNING (SFT)

To prepare Llama-3.1-8B-Instruct for RL, we first perform SFT priming as in (Gandhi et al., 2025).

**Data Generation:** We created a dataset of 1024 Crafter trajectories using Llama-3.3-70B-Instruct as a teacher. To ensure diversity, the teacher planned every $K$ steps ($K \sim U[2, 12]$ per trajectory) using 16 different planning prompts (Appendix A).

**SFT Priming:** The Llama-3.1-8B model was fine-tuned on this data, aligning the SFT process with the target RL configuration. For the **SFT+RL plan dynamically** agent, SFT targets use the dynamic format – `<plan>...</plan>` `[Action]` if the teacher planned at that step, otherwise just `[Action]` – along with a dynamic prompt encouraging this choice. Conversely, for the **SFT+RL no plan** agent, SFT targets include only the actions (`[Action]`, with all plan blocks removed), paired with a naive prompt focused solely on action prediction. This prepares the model appropriately for the subsequent RL phase. More details on the prompt in the Appendix A.

### 4.5 REINFORCEMENT LEARNING (RL)

We then used Proximal Policy Optimization (PPO) (Schulman et al., 2017) to fine-tune Llama-3.1-8B-Instruct agents in Crafter, optimizing task rewards possibly adjusted for planning costs (Sec. 3.3). We compare four key configurations:

- **Base+RL plan dynamically:** RL on the base model using a dynamic planning prompt.
- **Base+RL no plan:** RL on the base model using the naive (action-only) prompt.
- **SFT+RL plan dynamically:** RL on the SFT-primed dynamic planning model, using the dynamic planning prompt.
- **SFT+RL no plan:** RL on the naive SFT-primed naive model, using the naive prompt.

This isolates learned dynamic planning from fixed strategies, and the benefit of SFT priming with versus without explicit plan information. Further training details are in the Appendix C.

## 5 RESULTS

We present findings analyzing the impact of planning frequency in zero-shot settings and the effectiveness of our SFT priming approach, and the RL results.

### 5.1 ZERO-SHOT EVALUATION

To establish baseline capabilities and characterize the trade-offs between planning frequency, computational cost, and task performance, we conduct zero-shot evaluations. Specifically, we assess the zero-shot performance of Llama-3.3-70B-Instruct in the POGS and Crafter environments over 100 seeds. We systematically vary the agent's planning frequency, from never planning to planning at fixed intervals, and measured task progression against the mean number of output tokens (log scale), which serves as a proxy for the computational budget.

Contrary to the intuitive assumption that performance would scale monotonically with computational effort to form a Pareto frontier, our findings reveal a non-monotonic relationship, which we term the "Goldilocks" zone for planning frequency. As shown in Figure 1 (a-b), performance in both environments peaks at intermediate planning frequencies before declining as planning becomes more frequent. The degradation in performance associated with excessive planning aligns with the instability cost ($C_{noise}$) introduced in our conceptual framework (Section 3). Quantitative analysis of agent trajectories in the POGS environment (Appendix B) supports this view; the always-plan agent

**Human-provided plan 1**: "First, move east to fight the skeleton, then collect the iron ore."

**Human-provided plan 2**: "Now it's time to create a shelter: move east into the cave, then move west and place a stone block to seal yourself in."

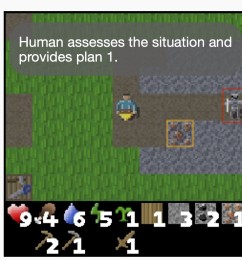
Human assesses the situation and provides plan 1.

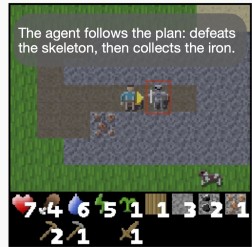
The agent follows the plan: defeats the skeleton, then collects the iron.

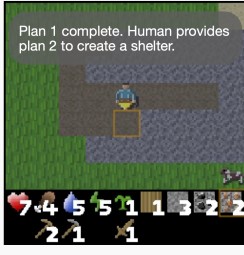
Plan 1 complete. Human provides plan 2 to create a shelter.

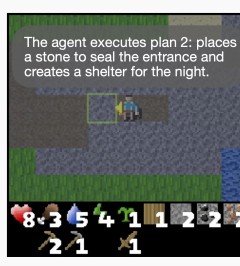
The agent executes plan 2: places a stone to seal the entrance and creates a shelter for the night.

Figure 3: **Human-Agent collaboration in Crafter**. We show an example where a human guides the agent with high-level plans to clear a cave from a skeleton, and create a shelter to survive the night, a complex behaviour that was not observed in any of the training runs otherwise.

exhibited the highest rate of backtracking, suggesting a tendency to oscillate or revisit states rather than explore efficiently. We hypothesize that this phenomenon may be compounded by other factors. For instance, excessive reasoning can lead to "overthinking" Sui et al. (2025) and longer contexts, which in turn dilute attention to the most critical information Liu et al. (2023).

Finally, it is noteworthy that a dynamic planning baseline is absent from our zero-shot analysis. Our initial attempts to elicit adaptive planning directly through complex prompting proved challenging and unreliable. Models struggled to consistently interpret abstract instructions such as "plan only when necessary," often defaulting to fixed patterns of always planning or never planning. This difficulty underscores the need for learning-based approaches, like SFT and RL, to effectively teach agents this meta-cognitive skill of deciding *when* to allocate test-time compute for planning.

## 5.2 SFT PRIMING

Having established in zero-shot evaluations that prompting alone fails to induce dynamic planning, we next turn to supervised fine-tuning. Recent work shows that RL tends to optimize within the existing behaviors of the model rather than induce new strategies Ma et al. (2025); Gandhi et al. (2025). This limitation underscores the necessity of an SFT priming stage that provides explicit demonstrations over a mixture of both planning and non-planning steps within a trajectory.

To evaluate the impact of SFT priming, we leverage the synthetic dataset introduced in Section 4.4. During this phase the models were trained either to predict both the explicit plans and corresponding actions ('Primed-Dynamic') or to predict only the actions ('Primed-Naive'). Importantly, both variants were derived from the same underlying action sequences, isolating the effect of explicit plans on imitation learning.

Figure 1c–d shows that model fine-tuned on the dynamic planning dataset ("Primed-Dynamic") achieve significantly higher task progression throughout training compared to model trained without plans ("Primed-Naive"). Moreover, the Primed-Dynamic agent exhibits lower KL divergence from the base model, suggesting that the inclusion of plans regularizes the fine-tuning process and prevents catastrophic forgetting. We hypothesize three complementary mechanisms underlying these improvements:

- **Explanatory Power:** Plans provide semantic rationales for actions, simplifying the behavioral cloning objective and making future actions more predictable.

- **Learning Planning Structure:** Exposure to explicit plan–action pairs encourages the model to internalize generalizable planning heuristics rather than memorizing isolated actions.

- **Regularization and Grounding:** Natural language plans act as a form of grounding signal, constraining the model's learning trajectory and reducing deviation from base capabilities.

Collectively, these findings indicate that explicit plan representations not only improve downstream RL sample efficiency but also enhance the SFT stage itself by stabilizing learning dynamics and accelerating task progression.

## 5.3 RL Fine-Tuning and Instruction Following

Following the supervised fine-tuning stage, we apply RL to further refine the agents' capabilities. The results (Fig. 1e-f and Fig. 3) from this phase highlight two main findings: (i) SFT priming is essential prerequisite for learning effective dynamic planning under RL; and (ii) the two-stage training process yields agents that can be steered by human-written plans to accomplish tasks beyond their autonomous performance.

Our experiments reveal that the SFT priming stage is crucial for enabling agents to develop dynamic planning skills through RL. The evidence for this is not just that SFT-primed agents outperform their non-primed counterparts, but is most starkly demonstrated by the fact that the Base+RL plan dynamically agent performed worse than the Base+RL no plan agent. This suggests that without the structured demonstrations provided by SFT, the base model cannot learn to plan effectively through RL alone; the attempt to generate plans is not only unhelpful but actively degrades performance. SFT priming is what provides the essential scaffolding, turning planning from a detrimental behavior into a highly effective strategy.

The agent primed with explicit plans (SFT+RL plan dynamically) significantly outperforms the agent primed only on actions (SFT+RL no plan). When planning is layered on top of SFT foundation, we observe improved sample efficiency and task progression. Qualitative analysis confirms that this agent successfully learns to generate and execute plans at varying levels of abstraction, demonstrating the ability to plan when necessary and replan in response to changing circumstances. We further observe that the planning cost penalty $C_{tokens}$ effectively regulates planning frequency and length in proportion to the penalty magnitude. Interestingly, higher penalties reduce explicit planning without overly degrading task performance (see Appendix C.3). This suggests that sufficiently trained agents begin to internalize planning behaviors, encoding necessary reasoning implicitly within the policy weights rather than relying on explicit token generation. Crucially, however, while internalized planning is efficient for autonomous performance, the capacity for *explicit* planning remains indispensable for interpretability and control. Without the structural interface of explicit plans, the agent cannot be effectively guided—a capability we demonstrate in the following section on human-AI collaboration.

**Steerability and Human-Agent Collaboration** While the training methodology produces significant performance improvements, computational constraints prevent any of the autonomous agents from fully solving the Crafter environment. To better understand the potential upper bound of our approach, we also evaluate agents in a human-in-the-loop setup.

The SFT+RL plan dynamically agent proves to be the most reliable at instruction following, making fewer mistakes during combat and adhering more consistently to the plans than the base and SFT-only models. Under human-provided plans, it successfully completes Crafter by collecting diamonds—an achievement not observed in autonomous runs (see Fig. 3 for a representative sequence). Additional qualitative examples and a Best-of-N analysis are provided in the Appendix D.

Table 1: Performance and efficiency comparison across training stages and model scales. Our fine-tuned 8B agent (SFT+RL) outperforms the 70B zero-shot baseline while generating significantly fewer tokens, and improves upon the SFT baseline in both reward and token efficiency.

| Method | Model | Planning Strategy | % of Max Reward | Avg. Output Tokens |
|---|---|---|---|---|
| Zero-shot | Llama-3.3-70B | Never | $0.343 \pm 0.012$ | 559.6 |
| Zero-shot | Llama-3.3-70B | Every 4 steps | $0.379 \pm 0.014$ | 11,510.9 |
| Zero-shot | Llama-3.3-70B | Always (Every step) | $0.349 \pm 0.013$ | 38,836.2 |
| SFT | Llama-3.1-8B | Never | 0.286 | 991.5 |
| SFT | Llama-3.1-8B | Every $k$ steps ($k \sim [2, 12]$) | 0.343 | 1,869.9 |
| Base+RL | Llama-3.1-8B | Never | 0.274 | 505.1 |
| Base+RL | Llama-3.1-8B | Plan dynamically | 0.210 | 10,818.7 |
| SFT+RL | Llama-3.1-8B | Never | 0.298 | 878.0 |
| **SFT+RL** | **Llama-3.1-8B** | **Plan dynamically** | **0.387** | **1,714.3** |

**Efficiency and Model Scaling.** To quantify the efficiency gains of our approach, Table 1 presents a direct comparison of task performance and token costs across model scales and training stages. Two key findings emerge. First, the SFT+RL dynamic agent is more efficient than its SFT precursor: it

achieves a significantly higher reward (0.387 vs. 0.343) while generating fewer tokens on average (1714.3 vs. 1869.9). Second, our pipeline enables a smaller model to outperform a much larger one. The Llama-3.1-8B agent, trained via our SFT+RL protocol, surpasses the peak performance of the Llama-3.3-70B zero-shot baseline (0.387 vs. 0.379). Crucially, it achieves this while being more efficient, reducing the number of output tokens 85% compared to the best-performing 70B configuration (1,714 vs 11,510 tokens).

**Adaptive Planning Efficacy and Plan Coherence.** To quantify the quality of learned dynamic planning, we used an LLM-as-a-judge approach (Gu et al., 2025) (see Appendix C.6) over 100 evaluation seeds. We assessed the Plan Completion Ratio (plans executed to completion within 15 timesteps) and the Adaptive Replans Ratio (plans abandoned and replaced due to justifying environmental shifts, such as encountering monsters or new resources). As shown in Table 2, the RL fine-tuning significantly improves the agent's planning loop. For the SFT+RL agent, 20% of generated plans are successfully brought to completion within 15 steps, and an additional 41% are appropriately discarded via adaptive replanning. The SFT+RL agent demonstrates a higher Plan Completion Ratio (0.20 vs. 0.16) and a higher Adaptive Replans Ratio (0.41 vs. 0.35) compared to the SFT-only agent. These improvements confirm that RL successfully refined the agent's decision policy ($\phi_\theta$) to more accurately assess when a plan has suffered plan drift and when replanning is necessary, and improving the cost-benefit trade-off of dynamic compute allocation.

Table 2: Consolidated Model Performance and Planning Statistics

| Method | Plan Completion | Adaptive Replans | Avg. Plan Distance | Median Plan Distance |
|--------|-----------------|------------------|--------------------|----------------------|
| SFT | $0.16 \pm 0.01$ | $0.35 \pm 0.01$ | $13.20 \pm 9.6$ | $12.0 \pm 1.4$ |
| SFT+RL | $0.20 \pm 0.01$ | $0.41 \pm 0.02$ | $23.29 \pm 23.3$ | $14.0 \pm 9.2$ |

## 6 DISCUSSION, LIMITATIONS & CONCLUSION

The ability for LLMs to leverage test-time compute has been transformative, yet efficiently allocating these resources in agentic settings stands as a critical, largely unexplored challenge. To the best of our knowledge, this work presents the first systematic investigation and learned solution enabling LLM agents to effectively allocate test-time compute in sequential decision-making tasks. Our findings reveal that prevailing always-plan (e.g. ReAct) and never-plan approaches are suboptimal. Instead, a "Goldilocks" effect emerges whereby intermediate frequencies outperform both extremes, likely avoiding instability ($C_{noise}$ in our framework) induced by excessive replanning. This highlights the need for LLM agents to strategically allocate, rather than naively scale, deliberation resources. Our two-stage SFT+RL methodology demonstrates that agents can learn this meta-cognitive skill, moving beyond fixed heuristics towards the adaptive, efficient behaviour essential for sustained autonomy. Moreover, the resulting agents become sufficiently adept at planning and execution to be effectively steered by human-written plans towards remarkable feats, including the full completion of Crafter through diamond collection, significantly impacting human-AI collaboration and safety.

While our results establish the value of dynamic test-time compute allocation, several limitations suggest directions for future work. Our experiments focused on specific models at certain scales (Llama-3.1-8B-Instruct for fine-tuning, Llama-3.3-70B-Instruct for evaluation) due to computational constraints. Investigating how optimal compute allocation strategies scale with model parameters would provide valuable insights. Additionally, extending this work beyond our current environments (POGS and Crafter) to more diverse domains would further validate the generality of our approach. Future research could also explore more sophisticated compute allocation mechanisms, scale up our experiments, or investigate methods to more explicitly integrate our conceptual framework's insights into novel RL algorithms.

In summary, this paper establishes that in zero-shot evaluation, per-timestep planning or reasoning strategies akin to ReAct are outperformed by "Goldilocks" planning frequencies, due to instability that results from excessive planning or "overthinking." By using a two-stage training methodology combining SFT and RL, we successfully train agents to dynamically allocate planning resources at test-time. This approach yields more effective and efficient behaviour compared to fixed planning strategies, marking a step towards more autonomous and scalable agentic systems.

## ETHICS STATEMENT

This work studies dynamic planning in language model agents within simulated environments, without human subjects or personal data. While improved planning could be misused in real-world autonomous systems, our evaluations are confined to sandboxed benchmarks, and we believe the scientific benefits outweigh these risks.

## REPRODUCIBILITY STATEMENT

We strive to make all experiments in this paper fully reproducible. We share the anonymous codebase at: https://anonymous.4open.science/r/LLM-Agents-Planning. We provide further descriptions of the training details in Appendix C.

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

# A  APPENDIX

## A.1  PLANNING PROMPTS

To ensure behavioural diversity in our SFT dataset, we designed 16 distinct planning prompts, each eliciting a different style of planning. These include prompts that ask the model to identify immediate subgoals, describe short-term action sequences, re-evaluate high-level strategies, or address gaps in information. The idea is to expose the model to a wide range of planning behaviours so that, during RL fine-tuning, it can learn when and how to use the most appropriate planning strategy depending on the situation. When planning is not required, a separate instruction prompt (Prompt 17) is used to guide the model to select the next action based solely on prior plans and observations.

---

### 1. Immediate-Subgoal Planner

```
Identify the immediate next subgoal required to progress towards the
overall task completion.
Outline your plan to achieve this specific subgoal, including any
necessary reasoning.

Output your plan strictly in the following format:
<plan>YOUR_PLAN_FOR_SUBGOAL </plan>
Replace YOUR_PLAN_FOR_SUBGOAL with your own plan.

Keep your plan relatively brief, only focusing on important
information.

After your plan, choose exactly ONE action from the allowed actions
list that initiates this plan.
Output no other text.
```

---

Prompt 1: Prompt encouraging subgoal identification and targeted execution.

---

### 2. Milestone-Focused Planner

```
Consider the overall objective.
What is the most crucial intermediate milestone to achieve next?
Explain why reaching this milestone is important for the overall task,
and outline the steps you'll take to get there.

Output your reasoning and plan strictly in the following format:
<plan>YOUR_REASONING_AND_SUBGOAL_PLAN</plan>
Replace YOUR_REASONING_AND_SUBGOAL_PLAN with your own plan.

Keep your plan relatively brief, only focusing on important
information.

After your plan, choose exactly ONE action from the allowed actions
list to start working towards this milestone.
Output no other text.
```

---

Prompt 2: Prompt focused on achieving the next key milestone and explaining its relevance.

### 3. Short-Term Sequence Planner

```
Detail the specific sequence of actions you intend to take over the
next few steps.
Explain the purpose of this sequence in relation to the current
situation.

Output your detailed short-term plan strictly in the following
format:
<plan>YOUR_DETAILED_SHORT_TERM_PLAN</plan>
Replace YOUR_DETAILED_SHORT_TERM_PLAN with your own plan.

Keep your plan relatively brief, only focusing on important
information.

After your plan, choose exactly ONE action from the allowed actions
list, which should be the first step in your detailed plan.
Output no other text.
```

Prompt 3: Prompt directing the agent to outline a short, purposeful action sequence.

### 4. Step-by-Step Immediate Planner

```
Think step-by-step for the immediate future.
What actions are needed right now and why?
Describe the logic connecting these immediate actions to the next
phase of the task.

Output your step-by-step thinking and plan strictly in the following
format:
<plan>YOUR_STEP_BY_STEP_LOGIC_AND_PLAN</plan>
Replace YOUR_STEP_BY_STEP_LOGIC_AND_PLAN with your own plan.

Keep your plan relatively brief, only focusing on important
information.

After your plan, choose exactly ONE action from the allowed actions
list that represents the very next concrete step.
Output no other text.
```

Prompt 4: Prompt guiding step-by-step reasoning for immediate next actions.

### 5. Gap-Bridging Phase Planner

```
Analyze the current state and the final goal.
Formulate a plan that bridges the gap, focusing on the most logical
next phase of work.
Explain how this phase contributes to the overall objective.

Output your analysis and plan strictly in the following format:
<plan>YOUR_BRIDGING_PLAN_AND_REASONING</plan>
Replace YOUR_BRIDGING_PLAN_AND_REASONING with your own plan.

Keep your plan relatively brief, only focusing on important
information.

After your plan, choose exactly ONE action from the allowed actions
list to begin executing this phase.
Output no other text.
```

Prompt 5: Prompt for bridging the gap between the current state and final objective.

### 6. High-Level Strategic Planner

```
Re-evaluate the overall strategy.
Outline your current high-level plan or strategic direction for
completing the task from this point forward, focusing on the major
phases ahead.

Output your strategic plan strictly in the following format:
<plan>YOUR_HIGH_LEVEL_STRATEGIC_PLAN</plan>
Replace YOUR_HIGH_LEVEL_STRATEGIC_PLAN with your own plan.

Keep your plan relatively brief, only focusing on important
information.

After your plan, choose exactly ONE action from the allowed actions
list that aligns with the first step of this strategy.
Output no other text.
```

Prompt 6: Prompt eliciting a broad, high-level plan covering major task phases.

### 7. Justified Approach Planner

```
Propose a plan for the next stage of the task.
Critically, justify *why* this sequence of steps (or this approach)
is the most sensible course of action right now.
Output your plan and justification strictly in the following format:

Output your plan and justification strictly in the following format:
<plan>YOUR_PLAN_WITH_JUSTIFICATION</plan>
Replace YOUR_PLAN_WITH_JUSTIFICATION with your own plan.

Keep your plan relatively brief, only focusing on important
information.

After your plan, choose exactly ONE action from the allowed actions
list that initiates your justified plan.
Output no other text.
```

Prompt 7: Prompt requiring justification for the chosen course of action.

### 8. Reasoning-Process Planner

```
Verbalize your thought process for deciding what to do next.
Explain your reasoning, considering the current situation and the
ultimate goal, and then state your resulting plan for the near term.

Output your reasoning process and plan strictly in the following
format:
<plan>YOUR_REASONING_PROCESS_AND_PLAN</plan>
Replace YOUR_REASONING_PROCESS_AND_PLAN with your own plan.

Keep your plan relatively brief, only focusing on important
information.

After your plan, choose exactly ONE action from the allowed actions
list based on your reasoning.
Output no other text.
```

Prompt 8: Prompt asking the agent to verbalize its reasoning before planning.

### 9. Approach-Comparison Planner

```
Briefly consider possible approaches for the next steps.
State the approach you choose to take and why it seems preferable to
alternatives right now.
Outline the plan based on this chosen approach.

Output your chosen approach, rationale, and plan strictly in the
following format:
<plan>YOUR_CHOSEN_APPROACH_RATIONALE_AND_PLAN</plan>
Replace YOUR_CHOSEN_APPROACH_RATIONALE_AND_PLAN with your own plan.

Keep your plan relatively brief, only focusing on important
information.

After your plan, choose exactly ONE action from the allowed actions
list that corresponds to your chosen approach.
Output no other text.
```

Prompt 9: Prompt comparing alternative approaches and selecting one.

### 10. Efficiency-Driven Planner

```
Devise a plan to make progress efficiently.
What is the most direct path to achieving the next significant step
or subgoal?
Outline this efficient path.

Output your efficiency-focused plan strictly in the following format:
<plan>YOUR_EFFICIENT_PLAN</plan>
Replace YOUR_EFFICIENT_PLAN with your own plan.

Keep your plan relatively brief, only focusing on important
information.

After your plan, choose exactly ONE action from the allowed actions
list that represents the first step on this path.
Output no other text.
```

Prompt 10: Prompt focused on generating the most direct, efficient plan forward.

## 11. Information-Gap Planner

```
Is there critical information missing?
If so, formulate a plan focused on gathering the necessary
information or resolving key uncertainties before proceeding with
the main task execution.
If not, state your plan for the next execution steps.

Output your information-gathering or execution plan strictly in the
following format:
<plan>YOUR_INFORMATION_OR_EXECUTION_PLAN</plan>
Replace YOUR_INFORMATION_OR_EXECUTION_PLAN with your own plan.

Keep your plan relatively brief, only focusing on important
information.

After your plan, choose exactly ONE action from the allowed actions
list relevant to this plan.
Output no other text.
```

Prompt 11: Prompt addressing missing information or uncertainty before acting.

## 12. Logical-Sequence Planner

```
Describe the logical sequence of operations you intend to perform
next.
Explain the dependency:  why does step B follow step A? Focus on the
immediate sequence.

Output your logical sequence and rationale strictly in the following
format:
<plan>YOUR_LOGICAL_SEQUENCE_PLAN</plan>
Replace YOUR_LOGICAL_SEQUENCE_PLAN with your own plan.

Keep your plan relatively brief, only focusing on important
information.

After your plan, choose exactly ONE action from the allowed actions
list representing the first operation in your sequence.
Output no other text.
```

Prompt 12: Prompt outlining a logically dependent sequence of actions.

---

**13. Short-term goal Planner**

```
Define your immediate goal for the next few actions.
Construct a plan specifically aimed at achieving this immediate goal.

Output your immediate goal and plan strictly in the following format:
<plan>YOUR_IMMEDIATE_GOAL_AND_PLAN</plan>
Replace YOUR_IMMEDIATE_GOAL_AND_PLAN with your own plan.

Keep your plan relatively brief, only focusing on important
information.

After your plan, choose exactly ONE action from the allowed actions
list that starts this plan.
Output no other text.
```

Prompt 13: Prompt focused on defining and achieving an immediate goal.

---

**14. Practical-Progress Planner**

```
Considering the available actions and the task objective, formulate a
practical plan for the next steps.
What needs to be done now to make steady progress?

Output your practical plan strictly in the following format:
<plan>YOUR_PRACTICAL_PROGRESS_PLAN</plan>
Replace YOUR_PRACTICAL_PROGRESS_PLAN with your own plan.

Keep your plan relatively brief, only focusing on important
information.

After your plan, choose exactly ONE action from the allowed actions
list to implement the first step.
Output no other text.
```

Prompt 14: Prompt aimed at formulating a grounded, actionable next step.

---

**15. Intent-and-Approach Planner**

```
State your intention for the next phase of action.
What do you aim to accomplish in the near future, and what's the
general approach?

Output your statement of intent and approach strictly in the
following format:
<plan>YOUR_INTENTION_AND_APPROACH</plan>
Replace YOUR_INTENTION_AND_APPROACH with your own plan.

Keep your plan relatively brief, only focusing on important
information.

After your plan, choose exactly ONE action from the allowed actions
list that reflects this intention.
Output no other text.
```

Prompt 15: Prompt stating the agent's intent and general method for proceeding.

**16. Next-Steps Planner**

```
Outline your plan for what to do next.
Keep it focused on the immediate steps required.

Output your plan strictly in the following format:
<plan>YOUR_NEXT_STEPS_PLAN</plan>
Replace YOUR_NEXT_STEPS_PLAN with your own plan.

Keep your plan relatively brief, only focusing on important
information.

After your plan, choose exactly ONE action from the allowed actions
list to start.
Output no other text.
```

Prompt 16: Prompt asking for a concise plan of immediate next actions.

**Act instruction**

```
Look at your previous plan and observations, then choose exactly ONE
action from the allowed actions listed previously.
Output no other text.
```

Prompt 17: Prompt instructing the model to directly choose the next action based on previous plan and observations.

## A.2 EVAL ZERO SHOT PROMPTS

Prompt 18 is used for the zero-shot evaluation agent that never plans and is simply tasked with outputting the next action based on previous observations. Prompt 19 is used for the `plan-every-k-steps` agents, where the model is instructed to generate a plan at fixed intervals and to act directly at other times. In such cases, Prompt 17 is used at steps where planning is not required. These zero-shot evaluations help assess the impact of planning frequency in the absence of fine-tuning.

---

**Never Plan Prompt**

```
Look at your previous observations, then choose exactly ONE action
from the allowed actions listed previously.
Output no other text.
```

---

Prompt 18: Prompt instructing the model to directly choose the next action based on previous observations.

---

**Plan Every K Prompt**

```
Review your previous observations and plan, then make a high-level
plan for completing the task.  Your plan can include reasoning about
how to solve the task.
After this planning phase, you will be asked to take actions one at a
time.

Output your plan strictly in the following format:
<plan>YOUR_PLAN</plan>
Replace YOUR_PLAN with your own thinking and plan.

After your plan, choose exactly ONE action from the allowed actions
listed previously.
Output no other text.
```

---

Prompt 19: Prompt used in fixed-frequency planning; elicits a plan at regular intervals.

## A.3 DYNAMIC PLANNING PROMPTS

Finally, Prompt 20 is used to replace all the aforementioned instruction prompts when creating the SFT dataset. This ensures the agent observes that the same prompt can elicit different planning strategies, with the goal that RL fine-tuning will enable it to learn to choose the appropriate type of plan. The adaptive nature of this prompt allows the model to autonomously decide when to create or update plans, facilitating flexible and context-aware decision-making.

**Dynamic Planning Prompt**

```
Review your current plan and observations.
• If you do not have a plan yet, create one.
• If your plan is outdated or needs changes, create a new plan.

If you create a new plan, output it in the following format:

<plan>YOUR NEW PLAN</plan>

Replace YOUR NEW PLAN with your revised plan.

If your current plan is still valid, proceed without outputting it
again.

After this evaluation (and any necessary replanning), output exactly
ONE allowed action.

Output nothing else except an optional <plan>...</plan> block and
that single action.
```

Prompt 20: Prompt allowing the model to decide when to plan based on task context.

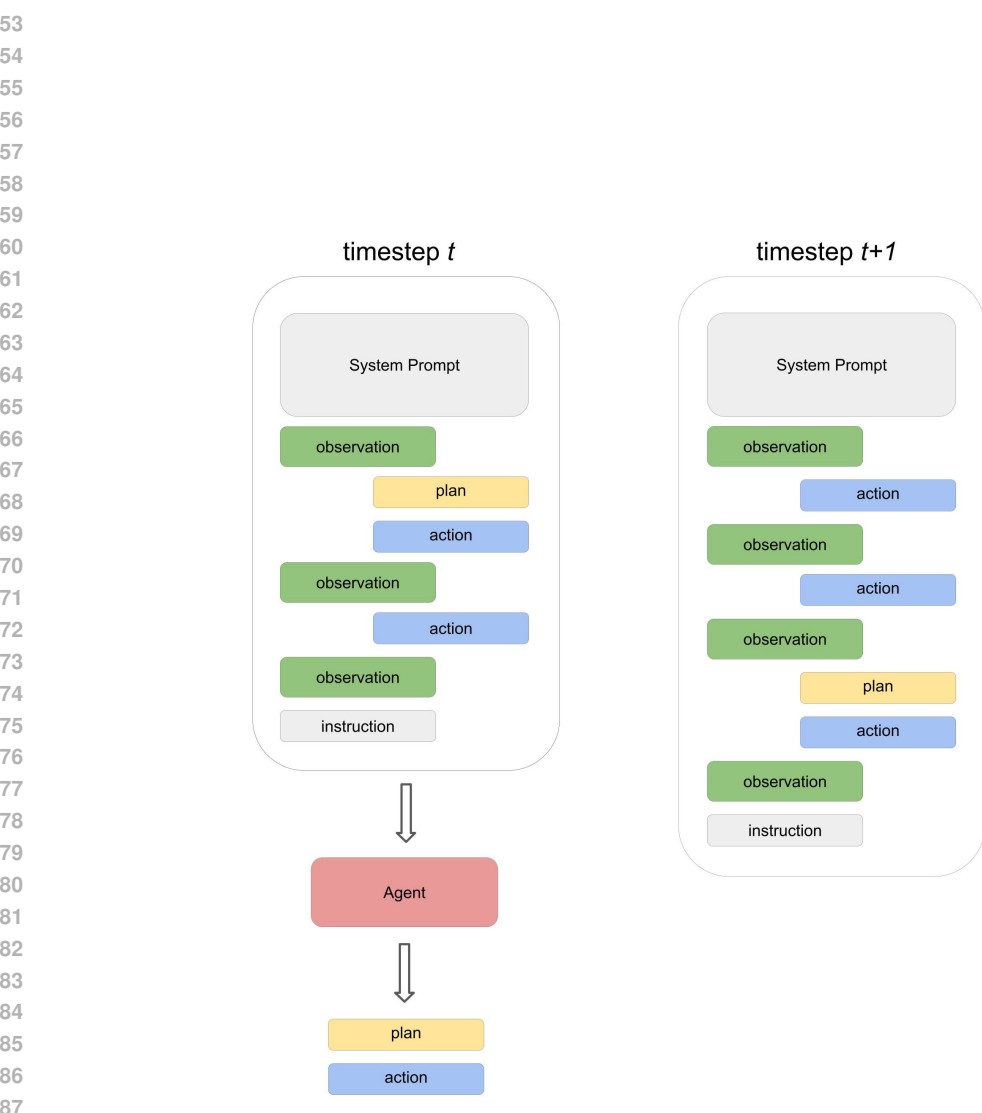

Figure 4: An illustration of the agent's input context over two timesteps, $t$ and $t + 1$. At each timestep, the agent processes a chat-formatted history composed of a system prompt, user messages (green `observation` and gray `instruction`), and assistant messages (yellow `plan` and blue `action`). The agent receives the history of interactions and in this case it generates a new plan and action. In the subsequent timestep $t + 1$, the input history is updated: the plan and action generated at $t$ are appended to the interaction history together with new observation, and the previous plan is removed. During experiments, to manage context length, the history provided to the agent was truncated to a maximum of 16 observations.

# B APPENDIX

## B.1 CRAFTER

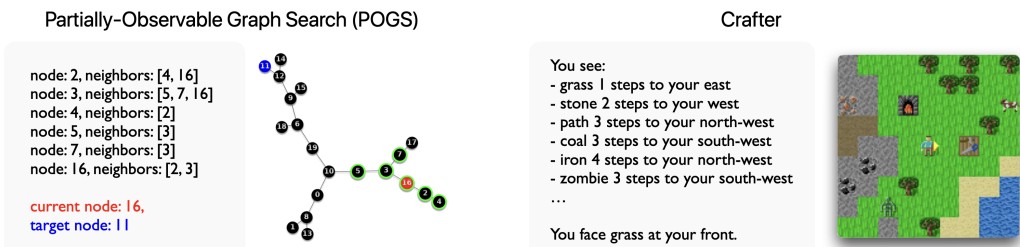

Figure 5: **POGS and Crafter environments.** POGS (left): Agent navigates a procedurally generated graph with partial visibility. Crafter (right): Agent receives natural language descriptions of terrain, resources, and creatures with their relative positions.

Crafter Hafner (2022) is an open-world survival game inspired by Minecraft. The environment is designed to benchmark reinforcement learning agents on tasks requiring generalization and long-term reasoning without extensive prior game knowledge. Crafter presents a procedurally generated 2D world where the agent must gather resources, craft tools, and defend against creatures to survive and unlock achievements. The environment provides observations as a combination of top-down pixel views of the agent's local surroundings and non-visual data, including the agent's health, inventory, and currently held item. The agent can perform 17 discrete actions, and its progress is measured by unlocking 22 predefined achievements, each yielding a sparse reward signal.

Our agent interacts with the Crafter environment through the BALROG Paglieri et al. (2025a) framework, which acts as a standardized wrapper. BALROG facilitates the communication between our agent, conceptualized as an underlying Large Language Model (LLM) combined with a specific prompting strategy, and the Crafter game environment. At each timestep, Crafter, via BALROG, relays the current observation to our agent. The agent, which internally maintains a history of past observations and actions, incorporates this new observation into its context. A dedicated prompt builder component within the agent updates this interaction history and formats it into a chat template. An example figure of Crafter together with current observation can be see in figure 5 on the right.

This prompt is then passed to the LLM, which processes the contextual information and generates the subsequent action as a natural language string. BALROG then translates this string into a game-compatible command for Crafter. Further details regarding the specifics of the system prompt, the wrapper's mechanics, and the handling of action validation can be found in BALROG Paglieri et al. (2025a).

## B.2 POGS

The Partially-Observable Graph Search (POGS) environment is a custom-designed, synthetic environment created specifically to isolate and evaluate planning under uncertainty. In POGS, agents navigate procedurally generated graphs, seeing only a limited area around their current node, and must find a path to a target node. This partial observability means agents often discover new sections of the graph or hit dead ends, forcing them to backtrack and adjust their plans. A key feature of POGS is its ability to quantify exploration efficiency by tracking backtracking, specifically defined as the number of times an agent visited the node it was on two steps prior; a lower backtracking count indicates better efficiency in solving the environment. To integrate POGS into the BALROG framework, we implemented a dedicated system prompt (see Prompt 21) that clearly specifies the agent's objectives, valid actions, and observation format, as demonstrated in the natural language observation example in Figure 5.

## B.3 BACKTRACKING ON POGS

Figure 6 visualizes additional metrics for POGS: success rate, length, backtrack count, and the number of output tokens. Increasing planning frequency leads not only to higher costs (more output tokens)

---

**The system prompt provided to the agent in POGS**

```
You are an AI agent designed to navigate the Partially Observable
Graph Search (POGS) environment.  Your primary objective is to find
and reach a specific target node.

The following are the only valid actions you can take in the game:
{list(range(env.num_nodes))}

In a moment I will present you with an observation containing:
- Adjacency list showing the neighbors of all currently visible nodes
- Your current node position
- The target node you need to reach

The graph has {env.num_nodes} nodes and is partially observable,
meaning you can only see connections within a k-nearest neighbor
radius of your current position.  In this episode k={env.k_nearest}.

Your action should be a single integer representing the label of the
node you want to travel to.  This node must be directly connected to
your current node.

PLAY
```

Prompt 21: The system prompt for POGS. This prompt guides the AI agent by defining its objective (to find and reach a specific target node in the POGS environment), outlining the valid actions (moving to an adjacent node), and detailing the structure of observations (adjacency list of visible nodes, current position, and target node).

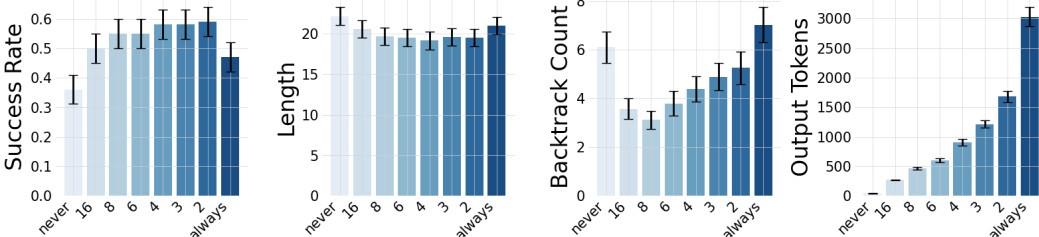

Figure 6: **Planning frequency affects exploration in POGS.** Intermediate planning frequency yields higher success rates and efficiency, while 'always-plan' agents show increased backtracking ($C_{noise}$).

but also to a higher backtrack count, with the 'always-plan' agent performing the most backtracking. Notably, this excessive backtracking correlates with a reduced success rate, as 'always-plan' agents exhibit lower performance compared to those planning less frequently. This suggests that planning too often causes the agent to continually change its mind, ultimately hindering its ability to efficiently navigate the graph and reach the target.

In contrast, agents that never plan also exhibit a high backtrack count, underscoring the value of having a plan. Planning appears to make the agent's behaviour more consistent and less erratic, which in turn leads to higher success rates. Overall, these results highlight the importance of balanced planning: planning too frequently or not at all hinders performance, while moderate planning improves efficiency and success.

## C  APPENDIX

### C.1  SUPERVISED FINE-TUNING

For the Supervised Fine-Tuning (SFT) phase, we generated a dataset of 1,024 trajectories using Llama 3.3 70B instruct. To ensure data diversity, we employed 16 distinct planning prompts (detailed in

Appendix A), which were sampled uniformly. The planning frequency for trajectories in the dataset was also sampled uniformly from the range [2, 12]. The Llama 3.1 8B instruct model served as the baseline for all SFT experiments, with no Low-Rank Adaptation (LoRA) adapters applied during this stage (Hu et al., 2022). We used AdamW (Loshchilov & Hutter, 2019) as the optimizer throughout.

We considered two objective functions: direct action prediction and full world modeling. Our findings, presented in Figures 7 and 8, indicate that omitting world modeling yields superior results in terms of both performance and reduced catastrophic forgetting. While we initially tested world modeling as a potential regularization technique to mitigate overfitting, it ultimately proved to be a distractor for the agents. To additionally model the environment dynamics, the weights of the model had to be changed more significantly, which is reflected in the higher KL divergence, and consequently led to increased forgetting and reduced model steerability. It is worth noting that for the agent to plan dynamically during SFT, it must also model the plans themselves, exposing it to a greater number of tokens during this stage. Training was performed using the DeepSpeed ZeRO Stage 3 optimizer (Aminabadi et al., 2022) to effectively manage memory and scale operations across multiple GPUs.

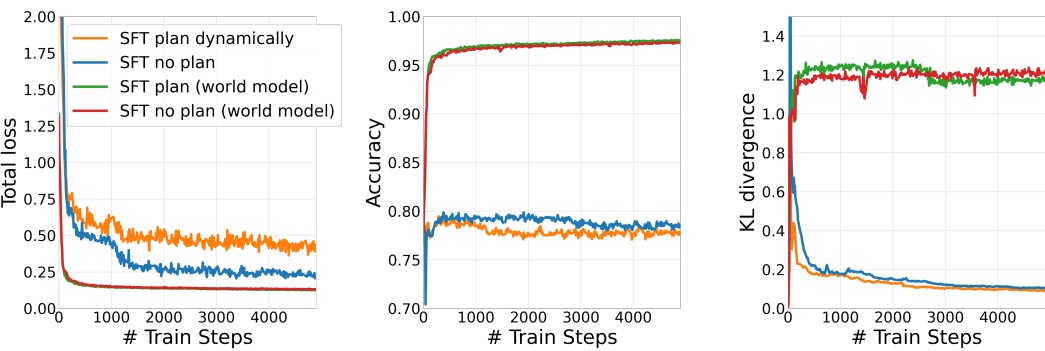

Figure 7: **SFT Training Metrics.** Comparison of (left) total loss, (center) accuracy, and (right) KL divergence across training steps for four SFT configurations. Models incorporating world modeling (green, red) exhibit lower training loss and higher accuracy but also show higher KL divergence. Among the configurations without world modeling, 'SFT plan dynamically' (orange) demonstrates highest total loss as modeling plans is much harder then modeling the world or just actions.

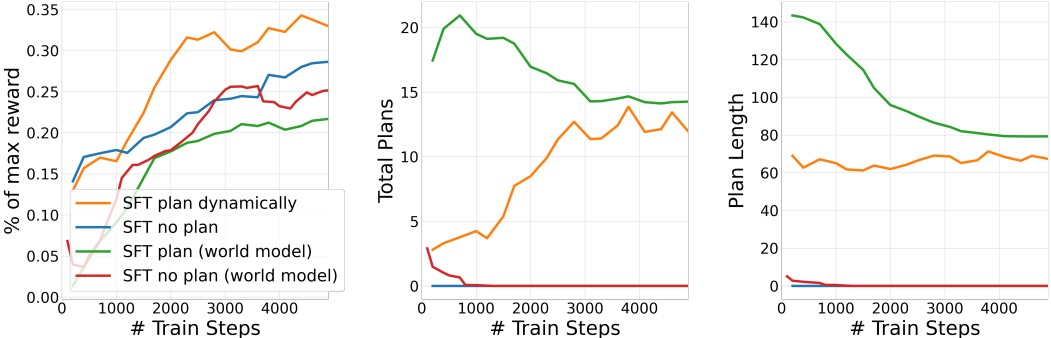

Figure 8: **SFT Evaluation Metrics.** Comparison of (left) normalized game score, (center) total number of plans generated, and (right) average plan length across training steps for the same four SFT configurations as in Figure 7. The configurations without world modeling (orange, blue) consistently achieve better task performance in terms of normalized score then configurations which additionally model the world (green, red).

The SFT hyperparameters are shown in Table 3.

### C.2   RLFT

During the Reinforcement Learning Fine-Tuning (RLFT) phase, we applied RSLoRA adapters (Kalajdzievski, 2023) separately to actor and critic models. Data collection involved using vLLM (Kwon et al., 2023) to host model weights and BALROG (Paglieri et al., 2025a) integrated with Ray (Moritz et al., 2018) to process outputs and handle agent-environment interactions efficiently. A

Table 3: SFT Training Configs

| Data & Model | |
| --- | --- |
| Model Size | 8B |
| Max Context Window | 8192 |
| LoRA | False |
| Dataset | 1024 trajectories |
| Optimization | |
| Learning Rate | 5e-6 |
| Beta (KL Coefficient) | 0.1 |
| Number of rollouts in batch | 384 |
| Rollout length | 16 |
| History length | 16 |

custom Proximal Policy Optimization (PPO) trainer (Schulman et al., 2017) was implemented, with model weights broadcasted back to vLLM after each training epoch. The actor model was based on the 8B parameter Llama 3.1 architecture, while the critic model utilized the 1B parameter Llama 3.3 architecture (Grattafiori et al., 2024). Similar to the SFT phase, the DeepSpeed ZeRO Stage 3 optimizer was employed for memory-efficient and scalable training.

Experiments were conducted using a total of 8 GPUs, allocating 6 GPUs for training and 2 for data collection. Most experiments were executed on a node with 8xH100 GPUs and typically completed within 24–48 hours. Initially, we experimented with reward penalties targeting invalid actions, excessively long responses, and overly frequent planning. However, such explicit reward shaping often resulted in agents refraining from planning entirely. Recognizing optimal planning frequencies ("Goldilocks zones"), such as planning every four steps rather than at every opportunity, we removed explicit reward shaping, thus allowing agents to autonomously learn optimal planning frequencies. As illustrated in Figure 9, agents trained under this strategy gradually improved their efficiency. Specifically, agents learned to execute plans less frequently (center panel) but with increased effectiveness, leading to shorter, more concise plans (right panel), and improved overall performance (left panel).

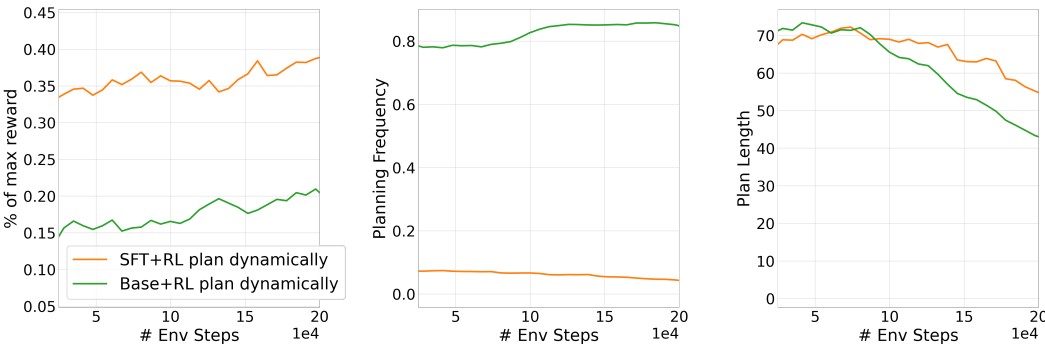

Figure 9: **Comparison of (left) Normalized Score, (center) Planning Frequency, and (right) Plan Length across environment steps for two RL configurations.** Both agents become more efficient with time; they learn to execute plans for longer, reflected in reduced planning frequency, and also generate more concise plans (reduced plan length).

Detailed hyperparameters for the RL experiments are provided in Table 4.

Table 4: RL Training Configs

| Data & Model | |
| --- | --- |
| Actor Size | 8B |
| Critic Size | 1B |
| Max Context Window | 8192 |
| Max output tokens | 200 |
| LoRA | True |
| **LoRA** | |
| R | 128 |
| Alpha | 64 |
| Dropout | 0.0 |
| **Optimization** | |
| Actor Learning Rate | 5e-6 |
| Critic Learning Rate | 5e-6 |
| Optimizer | AdamW |
| KL Coefficient | 0.05 |
| Number of rollouts in batch | 192 |
| Rollout length | 16 |
| History length | 16 |
| PPO Epochs | 1 |
| PPO policy clip | 0.2 |
| PPO value clip | 1.0 |
| Gamma | 1.0 |
| GAE | 0.95 |
| Value coefficient | 0.1 |
| Rollout Temperature | 1.0 |

## C.3 RL COST PENALTY ABLATION

We experimented with different penalties on the cost of planning ($C_{tokens}$) to analyze how agents adapt their behavior to computational constraints. As illustrated in Figure 10, the addition of these penalties led agents to reduce their planning frequency and plan length over time; however, we observed that the normalized score was largely unaffected regardless of penalty level. This is in contrast to our main findings, where explicit planning consistently enhances agent performance. This divergence suggests that, after sufficient training, agents may increasingly internalize planning behaviors; as they gain proficiency in the environment, much of the reasoning and strategy required for success can become implicit within the policy, reducing reliance on overt, explicit planning actions. This could help explain why further penalizing explicit plan generation produced only limited effects on final scores, even though explicit planning is crucial during earlier stages.

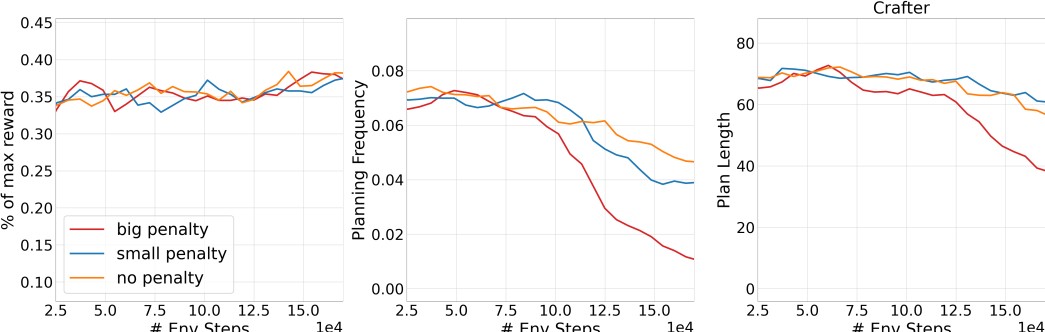

Figure 10: **Comparison of training with different planning cost penalties.** We compare the (left) Normalized Score, (center) Planning Frequency, and (right) Plan Length for agents trained with no penalty, a small penalty (-0.001 per token), and a big penalty (-0.005 per token).

## C.4 KL SENSITIVITY ANALYSIS

To evaluate the robustness of the SFT+RL pipeline and ensure the policy does not degenerate into trivial behaviors (always/never planning), we conducted a sensitivity analysis on the KL divergence coefficient ($\beta$) used in the PPO loss. We trained Llama-3.1-8B agents on Crafter for 200k environment steps, excluding learning rate warmup to explicitly test optimization stability. We swept $\beta \in \{0.05, 0.1, 0.15, 0.2, 0.3\}$.

The results, illustrated in Figure 11, demonstrate that the method is robust across intermediate values ($\beta \in [0.1, 0.2]$), showing very similar task progression. We identify $\beta = 0.1$ as the optimal configuration. We observed distinct failure modes at the extremes of the sweep. With insufficient regularization ($\beta = 0.05$), the policy tends to degenerate to a policy that only plans once at the beginning of the episode, akin to a knowledge dump. Conversely, with excessive regularization ($\beta = 0.3$), the model struggles to diverge sufficiently from the reference policy to optimize the action space, evidenced by a failure to reduce the frequency of invalid actions.

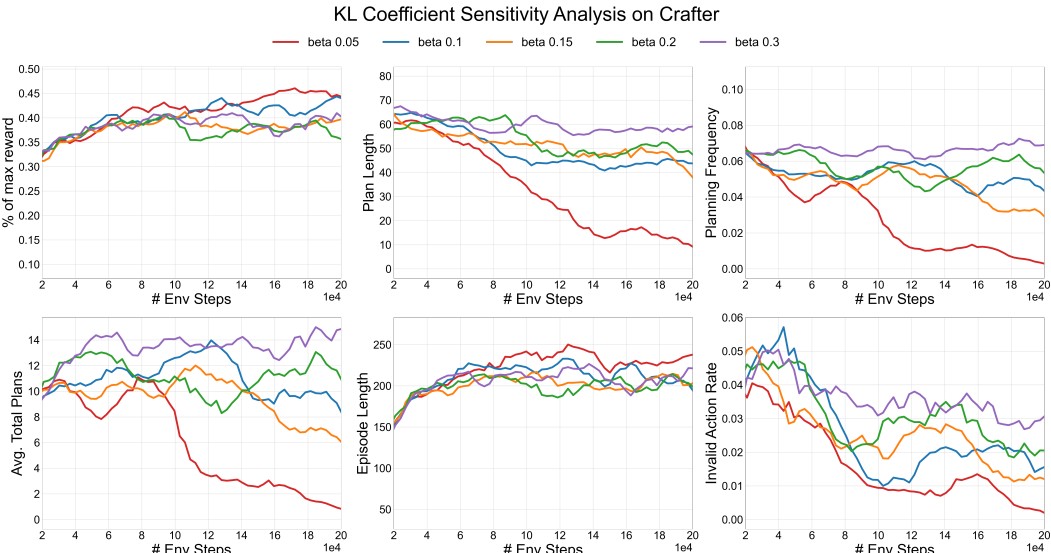

Figure 11: **KL Sensitivity Analysis on Crafter.** KL Sensitivity Analysis on Crafter. We compare task performance (% of max reward), plan length, planning frequency, total plans, episode length and invalid action rate across varying KL coefficients ($\beta \in \{0.05, 0.1, 0.15, 0.2, 0.3\}$) over 200k steps. The method proves robust across the $[0.1, 0.2]$ range. The plots reveal that $\beta = 0.1$ is a sweet spot; lower values ($\beta = 0.05$) lead to a collapse in planning frequency, while higher values ($\beta = 0.3$) invalid action rate high.

### C.5 ZERO SHOTS RESULTS

We ran further zero-shot experiments with Llama 3.1 8B, Llama 3.3 70B, and Gemini 2.5 Flash (Comanici et al., 2025) on Crafter, POGS, as well as the three TextWorld tasks used in BALROG (Côté et al., 2019; Paglieri et al., 2025a). Results can be seen in Figures 12, 13, and 14. "Goldilocks zones" can be observed across all three environments; however, their shapes may differ depending on the model tested. Llama 3.1 8B shows a sharp decline in performance the more it is allowed to plan on both Crafter and TextWorld, though never planning does not underperform as severely as it does in its 70B counterpart. On Gemini 2.5 Flash, always planning shows a marked degradation on Crafter, partly due to the instability cost ($C_{noise}$), and partly because the model sometimes confuses the `<plan>` token with the `<thinking>` as well as `<local_plan>` tokens that it is trained to use when scaling up test-time compute, despite reasoning being turned off during these experiments. We observe that Gemini 2.5 Flash's performance on POGS decreases with more plans compared to never planning and planning every 16 (which almost always leads to planning only once at the beginning). We hypothesize that this is due to POGS being a simpler environment that does not require any planning for stronger models like Gemini 2.5 Flash. In TextWorld, we observe that planning only brings moderate benefits compared to never planning, and results appear more noisy due to the higher difficulty and stochasticity of the environment. All experiments are run with 100 seeds, and error bars represent the standard error.

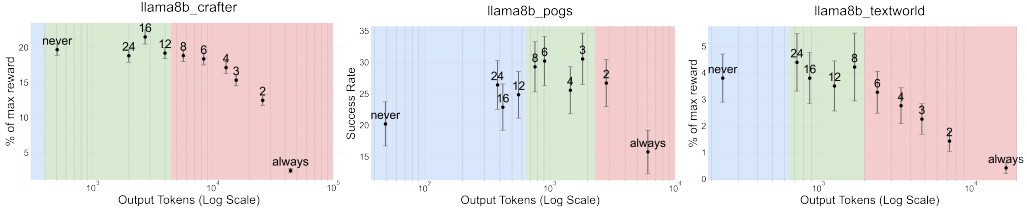

Figure 12: **Llama 3.1 8B zero shot results**.

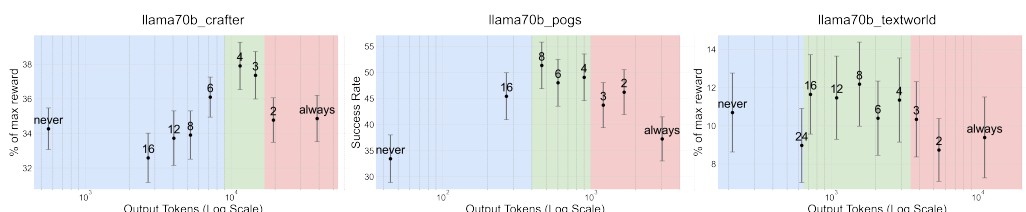

Figure 13: **Llama 3.3 70B zero shot results**.

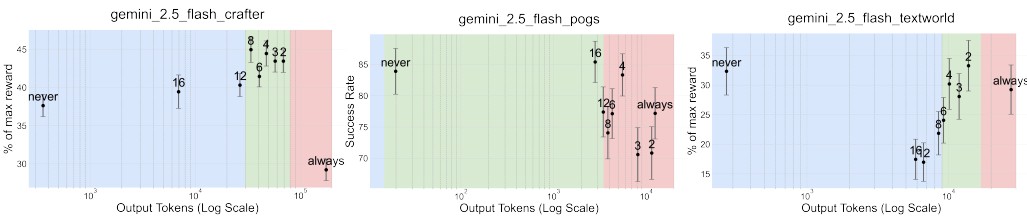

Figure 14: **Gemini 2.5 Flash zero shot results**.

## C.6 LLM-AS-A-JUDGE PROMPTS FOR PLANNING ANALYSIS

We employed GPT-5 mini (OpenAI, 2025) as a separate LLM as a judge to evaluate the quality of plan execution and replanning decisions for the planning statistics presented in Table 2. The judge analyzes trajectory segments, including the agent's observation history, the plan generated, and the subsequent actions. We used the following prompt structures for the two key metrics: Plan Completion and Adaptive Replanning.

---

**LLM as a Judge System Prompt**

```
You are a plan execution evaluator for an RL agent.  You will
analyze:
1.  An initial observation from the environment
2.  A plan generated by the agent
3.  A sequence of actions taken and their resulting observations.
Determine if the actions fully executed the original plan.
```

---

Prompt 22: System prompt for the LLM-as-a-judge.

---

**Plan Completion Query**

```
Based on the plan and execution history, has the plan been
successfully completed?  Think about it, then respond EXACTLY with
either:  <eval>YES</eval> or <eval>NO</eval>
```

---

Prompt 23: The query used to evaluate plan completion.

---

**Adaptive Replanning Justification Query**

```
You are verifying whether proposing a new plan was justified.
Analyze the observation history up to the current plan proposal.
Respond YES if either:  1.  The previous plan was completed before
this plan started, or 2.  The new plan is clearly motivated by
unexpected circumstances (new threat, low health/hunger/energy, etc.).
Otherwise respond NO.
```

---

Prompt 24: The query used to evaluate adaptive replanning.

### C.6.1 PLANNING DISTANCE STATISTICS

The statistics below, corresponding to the "Plan Distance" columns in Table 2, show the number of steps an agent executes under a given plan before a decision is made to either complete or replace that plan. The increase in the \*\*Average Plan Distance\*\* for the SFT+RL agent (23.29 vs. 13.20) suggests that the RL agent successfully learns to maintain plans for longer periods. Crucially, the \*\*Median Plan Distance\*\* remains similar (14.0 vs. 12.0), indicating that the higher average is driven by more outliers, where the SFT+RL agent correctly avoids replanning for extended, low-uncertainty sequences.

Table 5: Consolidated Model Performance and Planning Statistics. **Plan Distance** refers to the number of steps taken before the agent either successfully completed the plan or initiated a replanning decision.

| Method | Plan Completion | Adaptive Replans | Avg. Plan Distance | Median Plan Distance |
|--------|-----------------|------------------|--------------------|----------------------|
| SFT | $0.16 \pm 0.01$ | $0.35 \pm 0.01$ | $13.20 \pm 9.6$ | $12.0 \pm 1.4$ |
| SFT+RL | $0.20 \pm 0.01$ | $0.41 \pm 0.02$ | $23.29 \pm 23.3$ | $14.0 \pm 9.2$ |

# D APPENDIX

## D.1 QUALITATIVE RESULTS

In this section, we present further qualitative examples to illustrate the capabilities and limitations of our SFT+RL dynamically planning agent.

First, Figure 15 showcases successful human-agent collaboration, detailing how human-provided high-level plans guided the agent to a game-winning Crafter trajectory after approximately 20 attempts. Next, we demonstrate its autonomous dynamic planning. Figure 16 shows the agent interrupting an ongoing task and adaptively replanning to acquire critical food supplies when its health is low. Figure 17 further illustrates its dynamic planning capabilities, observing the agent employing multi-stage tactics: initially planning to craft a weapon, then devising a new plan to strategically position itself before engaging enemies. Finally, Figure 18 presents an execution failure, showing how an otherwise sound plan to craft an item falters because the agent fails to verify all necessary prerequisites—specifically, by not ensuring a required furnace is accessible. This is akin to the knowing-doing gap identified in BALROG (Paglieri et al., 2025a).

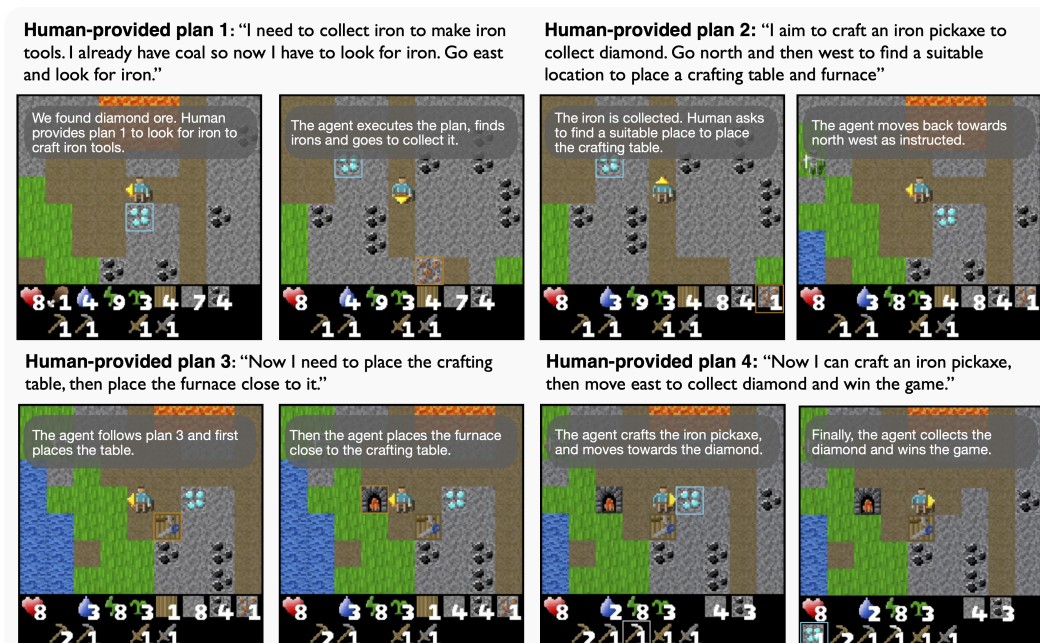

Figure 15: **Human-Agent Collaboration in Crafter**. This figure illustrates a successful human-agent collaboration, where human-provided plans guided the RL-trained planning agent to complete the game by mining diamond.

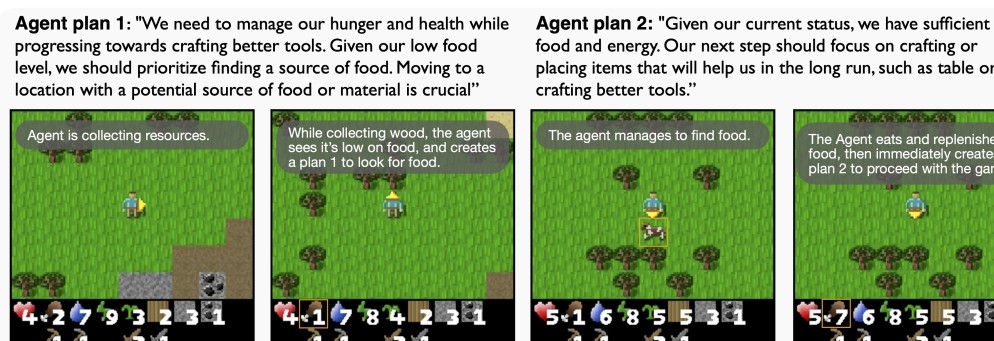

Figure 16: **Autonomous Replanning for Survival.** The SFT+RL plan dynamically agent demonstrates adaptive behavior by interrupting its current objective to address a critical need. It formulates a plan to acquire food when low on health, and upon replenishing, generates a new plan to resume game progression.

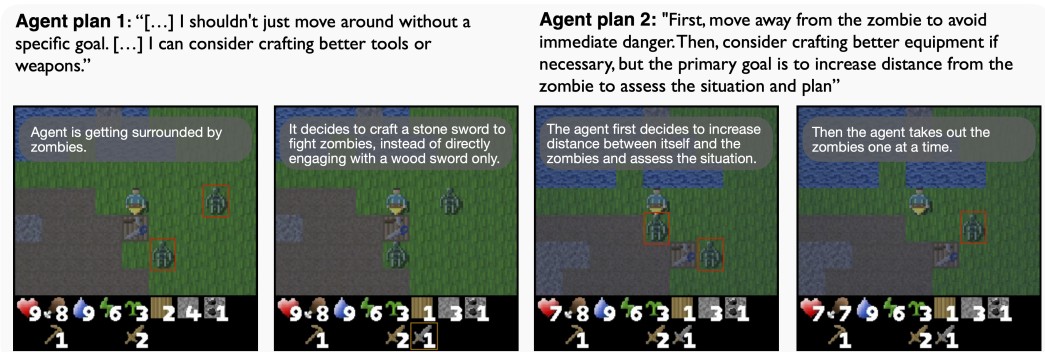

Figure 17: **Autonomous Multi-Stage Tactical Planning.** The agent showcases its ability to chain plans and adapt its tactics. After initially planning and crafting a stone sword to combat zombies, it reassesses and creates a subsequent plan to first gain distance, demonstrating a more strategic approach before engaging.

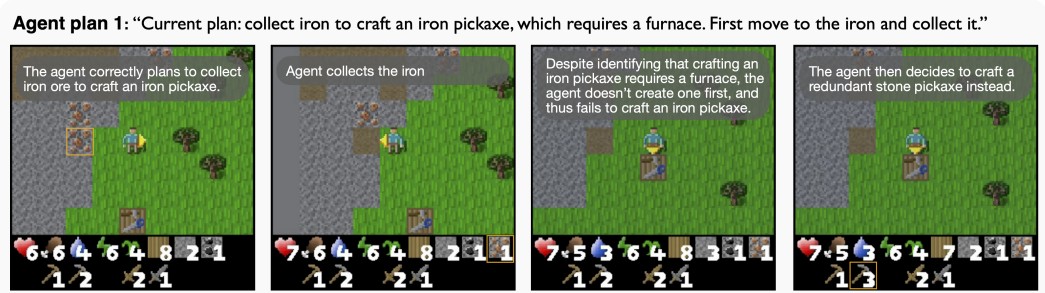

Figure 18: **Agent Failure Case**. The agent correctly plans to craft an iron pickaxe, identifying the necessary iron and furnace. However, after successfully collecting iron, it fails by attempting the craft action without ensuring the furnace is accessible, indicating a lapse in verifying all conditions of its plan.

## D.2 BEST-OF-N UPPER BOUND

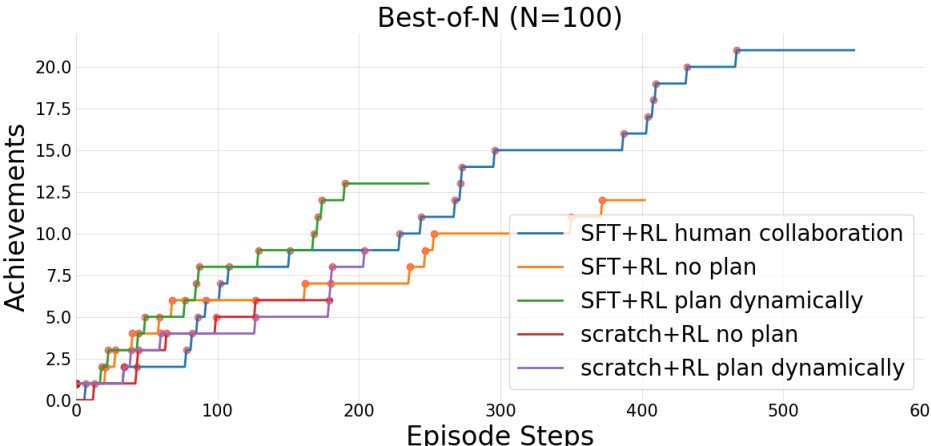

Figure 19: **Best-of-N (N=100) comparison on Crafter.** The plot shows achievements versus episode steps for each method, evaluated on the best of N trajectories. Human collaboration (N = 20) demonstrates the strongest progression, followed by SFT+RL plan dynamically and SFT+RL no plan, with base RL baselines lagging behind.

To quantify the upper bound of performance, we conducted a Best-of-N analysis, comparing the best runs attained by each method across 100 independent runs, and 20 runs for the human collaboration. As shown in Figure 19, the SFT+RL when guided by human plans, successfully solves Crafter by mining a diamond, significantly outperforming all of the methods. This result demonstrates the impact of human steering and further validates the benefits of conditioning the models with plans.

