# OpenReview forum: "Learning When to Plan: Efficiently Allocating Test-Time Compute for LLM Agents"
_ICLR.cc/2026/Conference — Submitted to ICLR 2026_

### Official Review · Reviewer_Fq61 · 2025-10-29

**Soundness:** 3
**Presentation:** 3
**Contribution:** 2
**Rating:** 4
**Confidence:** 5

**Summary:**

The paper addresses a critical limitation of existing LLM agents in sequential decision-making: extreme planning strategies (either "always planning" like ReAct or "never planning") lead to high computational costs/performance degradation or limited performance, respectively. To solve this, it proposes a conceptual framework for dynamic planning that formalizes the cost-benefit trade-offs of planning (balancing performance gains, computational costs, and behavioral instability).

The authors design a two-stage training pipeline to teach LLM agents when to allocate test-time compute for planning: (1) Supervised Fine-Tuning (SFT) on diverse synthetic trajectories (with 16 planning prompt styles) to prime models with planning behaviors; (2) Reinforcement Learning (RL) (via PPO) to refine dynamic planning in long-horizon environments.

Experiments on two environments (POGS, a partially observable graph search; Crafter, a Minecraft-inspired grid-world) show four key results: (1) Each task has a "Goldilocks" planning frequency (intermediate, outperforming extremes); (2) SFT with explicit plans improves imitation learning and reduces catastrophic forgetting; (3) SFT+RL yields agents with better sample efficiency that plan/replan adaptively; (4) Post-RL agents can be steered by human plans to achieve feats (e.g., collecting diamonds in Crafter) beyond autonomous capabilities.

**Strengths:**

1. An incremental exploration of dynamic test-time compute for LLM Agents: Breaking through the extreme strategies of "always planning / never planning" and proposing a formalized cost-benefit framework;

2. Two-Stage Training Pipeline: Injecting planning bias through SFT (Supervised Fine-Tuning) and optimizing dynamic strategies via RL (Reinforcement Learning), balancing efficiency and performance;

3. Potential of Human-AI Collaboration: Proving that the planning agent after RL can be guided by humans, enhancing controllability and security.

**Weaknesses:**

1. Overclaim: The authors assert that “this work is the first to explore training LLM agents for dynamic test-time compute allocation in sequential decision-making tasks.” However, this claim is inaccurate. For instance, Hu et al. (2024) have already investigated the same problem in the following work
@article{Hu2024,
  title={Enabling intelligent interactions between an agent and an LLM: A reinforcement learning approach},
  author={Hu, Bin and Zhao, Chenyang and Zhang, Pu and Zhou, Zihao and Yang, Yuanhang and Xu, Zenglin and Liu, Bin},
  journal={Reinforcement Learning Journal},
  year={2024}
}

Indeed, several findings reported in the current submission appear consistent with or have been previously revealed in Hu et al. (2024). A more comprehensive literature review is therefore recommended, particularly one that explicitly clarifies the relationship and differences between this work and Hu et al. (2024).

2. Model Scale Limitations: Only tested Llama-3.1-8B (fine-tuned) and Llama-3.3-70B (zero-shot), without verifying the scaling effects of larger models;

3. Environmental Scope Limitations: Experiments were only conducted in POGS and Crafter. The scope needs to be expanded to scenarios such as robot control and real-world decision-making;

4. Simplified Planning Mechanism: More complex planning strategies (e.g., hierarchical planning, multimodal planning) have not been explored.

**Questions:**

See the weakness Section.

---

> ### Author Response · Authors · 2025-11-25
>
> We thank the Reviewer Fq61 for their thoughtful and detailed comments. We appreciate the acknowledgment of our formalized cost-benefit framework, two-stage training pipeline, and the potential for human-AI collaboration.
>
> ### **Novelty and comparison to previous work**
> Regarding the concern about the novelty of our work compared to Hu et al. (2024), we agree that our original statement claiming to be the "first" to explore this area was overly broad, and we have removed the absolute language. However, we respectfully point out a fundamental distinction between our approaches. Hu et al. (2024) train a separate "Mediator" policy to decide when to query a fixed LLM planner, effectively treating planning as an external tool use problem. In contrast, our work focuses on finetuning a single, monolithic LLM to implicitly learn when to allocate its own test-time compute by predicting a special <plan> token within its unified output stream. This is a crucial step toward models that internalize the reasoning-action loop rather than relying on external modular components. We have updated our Related Work section to explicitly discuss Hu et al. (2024) and clarify this distinction between external "asking policies" and our internal dynamic compute allocation.
>
> ### **Model scale and environmental scope**
>
> Regarding the concerns about model scale and environmental scope, we acknowledge that our fine-tuning experiments were limited to Llama-3.1-8B and the POGS/Crafter environments due to the high computational costs of RL fine-tuning. However, we believe our results still offer robust evidence for our core claims. To address the concern about scaling effects, we highlight that our zero-shot evaluations were performed on the much larger Llama-3.3-70B model. These experiments confirmed that the non-monotonic "Goldilocks" relationship between planning frequency and performance persists at larger scales, suggesting that the phenomenon we investigate is not an artifact of smaller models. Furthermore, while we did not test on robot control, Crafter was specifically chosen because it serves as a rigorous benchmark for the many challenges found in real world tasks, namely, high-dimensional partial observability and the requirement for long-horizon survival and resource management strategies. We believe these environments sufficiently isolate the decision-making challenges relevant to dynamic planning.
>
> ### **On the simplified ReAct planning mechanism**
> Finally, regarding the observation that we utilized a simplified planning mechanism, we agree that we focused on single-step, ReAct-style reasoning within a unified output format. While extending this to hierarchical or multimodal planning is an exciting avenue for future research, our current work aims to establish the fundamental mechanism of when to deliberate. We view this as a necessary precursor to the question of how to deliberate using more complex modules. By successfully demonstrating that an agent can learn to gate its reasoning based on a state-dependent cost-benefit analysis, we provide a foundational step that future work can build upon to trigger more sophisticated planning subroutines.
>
> We hope that these clarifications, particularly the explicit differentiation from similar prior work and the justification of our experimental scope, address your concerns and motivate you to increase your support for our paper.

---

### Official Review · Reviewer_dRJY · 2025-10-30

**Soundness:** 3
**Presentation:** 2
**Contribution:** 3
**Rating:** 2
**Confidence:** 4

**Summary:**

The work presents a framework for training LLM agents to dynamically allocate test-time compute for planning. The framework is a 2 stage system, where the model is first SFT on synthetic data to seed the model for dynamic planning and then RL is applied to refine and enhance this capability to perform the best in long-horizon environments while reducing costs. Using this framework, the authors show that this setup can train dynamically planning agents which are simultaneously sample efficient and able to perform better on complex tasks.

**Strengths:**

The methodology section of the paper is quite thorough and provides a detailed breakdown of their dynamic planning architecture. The problem statement is formalized carefully and the authors do a good job to break down different aspects of the costs related to planning, as well as the dynamics of planning in Sections 3.1 and 3.2. The flow of the results section in Section 5 is quite good as the authors show a "planning Pareto frontier" and use this as motivation for their results in Section 5.2 and 5.3. Results show that their method outperforms baselines.

**Weaknesses:**

The main weakness is a limited number of datasets tests and a lack of experimental results overall. The main experimental results in this paper are completely contained with Figure 1, and the authors only show their main SFT and RL results using the Crafter dataset. Results would be significantly stronger if there was more than just 1 dataset and also 1 base model evaluated using their framework.

Furthermore, there is no direct comparison between the RL trained model and the pure SFT based model. There is not a single table anywhere in the paper where the actual comparison between zero-shot, SFT, and SFT + RL is compared, which make it very difficult to evaluate how well this framework performs compare to pure SFT and prompting. For instance, in Figure 1a, it seems like a naive prompted model which plans every 4 steps achieves the same performance as the SFT based model, which looks concerning as I would expect SFT model to perform better. Figure 1a also uses "Normalized Score" on the Y axis, where as Figure 1c and 1e use "\% of max reward", are these the same metrics for Crafter?

There is also not a single result looking at total number of tokens generated for plans or actions in the main paper. Figure 1e and 1f only show performance and planning frequency, but one of the benefits of their framework, theoretically, is reduced costs from getting rid of superfluous planning when it is unnecessary. Analysis would be significantly strengthened with some kind of results or table detailing these results.

Furthermore, in section 5.3, authors state that:

"We further observe that the planning cost penalty C tokens introduced in Section 3.3 effectively adjusts agent behavior
in proportion to the penalty magnitude, with higher costs leading to reduced planning frequency
and length during training while preserving task performance, indicating that agents can flexibly
adapt their planning strategies to different computational constraints" [lines 439-443]

However, isn't this cost only used during RL Training? Once you actually deploy the model, it is not like if you change the cost of planning, suddenly the model will change its behavior.

In addition, authors claim to be the first work to explore training LLM agents for dynamic test-time compute allocation for planning; however, one prior work, UI-Tars [1] seems to be the first work to explore training LLMs for dynamic planning. In [1], Section 4.4, they use ActRe to collect trajectories to train the model to perform planning and reflection dynamically using SFT. At runtime, the model dynamically decides to use System-2 reasoning during inference without being prompted.

Section 3.2 on Plan Drift not mentioned or referenced anywhere else in the paper. The authors seem to suggest a taxonomy for planning related errors, but there is no analysis or experiments related to this.

[1] Qin, Yujia, et al. "Ui-tars: Pioneering automated gui interaction with native agents." arXiv preprint arXiv:2501.12326 (2025).

**Questions:**

Can the authors use their framework on any additional datasets or any additional models to show that their result generalize?

Furthermore, can the authors provide a clear numerical table showing the Zero-shot Prompted, SFT (and variants), and RL + SFT (and variants) performance on these datasets?

It would also be very interesting to see an analysis of the # of tokens generated, as well as the prompting frequency for each of the different baselines and variants. Does this RL + SFT method successfully reduce the cost of running the model?

In Figure 1f, it seems to show that the best model seems to plans every 10?-ish steps? Does a naive prompted model that is prompted to plan every 10-ish steps perform worse?

---

> ### Author Response · Authors · 2025-11-25
>
> We thank Reviewer dRJY for their constructive feedback and for recognizing the thoroughness of our methodology and the clarity of our problem formalization. We appreciate that you found the flow of our results section effective and that our method clearly outperforms baselines. We have carefully addressed your concerns regarding experimental scope, baselines, and cost analysis through the clarifications and additional data detailed below.
>
> ### **Model scale and performance comparisons**
> You raised a concern regarding the comparison between the zero-shot results in Figure 1a and the SFT results, noting that a naive prompted model planning every 4 steps appeared to achieve similar performance to the learned SFT model. We would like to clarify that the naive zero-shot results in Figure 1a were generated using Llama-3.3-70B, while our SFT and RL results were achieved using the much smaller Llama-3.1-8B. The fact that our 8B model matches or exceeds the performance of a 70B model proves the efficiency of our training pipeline. Regarding your question on metrics, we confirm that "Normalized Score" and "% of max reward" refer to the same metric across all Crafter figures; we have unified the axis labels in the revision to avoid confusion.
>
> ### **Efficiency analysis**
> To address your request for a direct numerical comparison and an analysis of token costs, we have compiled a comprehensive comparison table (Table 1, provided below) uniting these metrics. This data highlights the efficiency gains of our approach. As you correctly noted, a key theoretical benefit of our framework is cost reduction. The data confirms this: our SFT+RL (Plan dynamically) agent achieves a significantly higher reward (0.387) compared to the SFT baseline (0.343) while generating fewer tokens on average (1714.3 vs. 1869.9). This empirically validates that the agent has learned to allocate compute more efficiently, planning only when necessary rather than at every step or fixed intervals.
>
> | Method | Model | Planning | % of max reward | Avg. Output Tokens |
> | :--- | :--- | :--- | :--- | :--- |
> | Zero-shot | Llama-3.3-70B | Never | 0.343 ± 0.012 | 559.6 |
> | Zero-shot | Llama-3.3-70B | Every 4 steps | 0.379 ± 0.014 | 11510.9 |
> | Zero-shot | Llama-3.3-70B | Always (Every step) | 0.349 ± 0.013 | 38836.2 |
> | SFT | Llama-3.1-8B | Never | 0.286 | 991.5 |
> | SFT | Llama-3.1-8B | Every k steps; k~[2,12] | 0.343 | 1869.9 |
> | Base+RL | Llama-3.1-8B | Never | 0.274 | 505.1 |
> | Base+RL | Llama-3.1-8B | Plan dynamically | 0.210 | 10818.7 |
> | SFT+RL | Llama-3.1-8B | Never | 0.298 | 878.0 |
> | SFT+RL | Llama-3.1-8B | Plan dynamically | 0.387 | 1714.3 |
>
> ### **Cost penalty in training**
> Regarding your question about the cost penalty mechanism, you asked if the cost is only used during training and how it affects deployment. You are correct that the penalty $C_{tokens​}$ is part of the reward function used only during RL training. The mechanism works by shaping the agent's policy parameters. During training, the agent learns that generating excessive tokens results in a lower return due to the penalty. Consequently, the deployed policy internalizes this trade-off, naturally opting to generate fewer tokens or plan less frequently at inference time without needing an external penalty signal. This is evidenced by our ablation study in Appendix C.3, where agents trained with higher penalties learned to produce shorter plans and plan less frequently.

---

> ### Author Response · Authors · 2025-11-25
>
> ### **Comparison with UI-Tars**
> Regarding novelty and the comparison to UI-Tars, we appreciate you highlighting this concurrent work [1]. While UI-Tars is indeed a valuable contribution that learns to execute plans dynamically via SFT on collected trajectories, our work specifically targets the explicit optimization of test-time compute allocation via RL. Our method does not just imitate a dynamic teacher; it learns when planning is worth the cost. We have clarified this distinction and cited UI-Tars in the related work section. Similarly, regarding "Plan Drift," we clarify that this concept (Section 3.2) serves as the theoretical motivation for why static planning schedules fail (motivating the "Goldilocks" zone), rather than an experimental variable to be optimized directly.
>
> ### **Generalization of goldilocks zone**
> We also agree that testing on additional environments would strengthen the paper. While computational constraints prevent running the full SFT+RL pipeline on entirely new domains (addressing your question 1), the validity of our method rests on the premise of the "Goldilocks zone." To validate this, we have conducted additional zero-shot experiments on TextWorld (three environments taken from Balrog, similar in style to ALFworld) using Llama-3.3-70B and Gemini-2.5-Flash. Results can be seen in the new appendix C.5 of the paper. We observed the same non-monotonic relationship between planning frequency and performance that we reported in Crafter and POGS on Llama 3.3 70B. This confirms that the "Goldilocks" phenomenon is generalizable across many sequential decision-making tasks. We have also clarified in the paper that our method is specifically designed for these types of sequential tasks; for single-turn domains like mathematical reasoning where "always planning" (Chain-of-Thought) is typically optimal, our dynamic allocation method would naturally converge to always planning and thus bring fewer benefits.
>
> ### **Fixed frequency baseline**
> Finally, regarding your suggestion about the fixed-frequency baseline in Figure 1f, we acknowledge that a direct comparison to a naive model prompted to plan exactly every 10 steps (matching the learned frequency) would add valuable context. We are currently finalizing this experiment to isolate the specific benefit of dynamic allocation versus static frequency at the same average cost. We will include these results in the final version of the paper.
>
> With the clarification on model sizes (8B vs 70B) and the new efficiency data in Table 1, we believe we have addressed your main reasons for rejection and hope you will reconsider your rating.

---

> > ### Comment · Reviewer_dRJY · 2025-11-26
> >
> > I thank the authors for their responses.
> >
> > I like the efficiency analysis in Table 1, which makes it much more clear how your method improved performance over the baselines. To improve clarity, I might add that the authors should add an effective Planning number with the mean and standard deviation of how often the RL trained models tend to replan.
> >
> > With regards to the Cost penalty, my issue was that the writing seemed to incorrectly suggest that planning cost could be dynamically adjusted during test time to scale the efficiency of the model up or down, but it seems like this statement has been changed in the paper.
> >
> > Regarding the analysis of the Goldilocks zone, it is good that the authors have done additional experiments on further environments, zero-shot, to show generalization. However, it still doesn't address my concerns about the lack of experiments on additional environments.
> >
> > Figure 1 is still a bit messy, and you even have the red bar for typos in the figure, so I would do a thorough cleanup of the manuscript to clean up these typos.
> >
> > Overall, I have raised my score to a 4, but I do not believe that I can raise the score to a 6 without at least 1 of the following:
> > 1. As I asked originally, just 1 additional experiment in a different domain in order to show that the method generalizes beyond Crafters.
> > 2. In lieu of additional experiments which may be too costly, I would like to see an in depth analysis on the planning behavior of the final model itself. Right now, the paper states:
> > >When planning is layered on top of SFT foundation, we
> > observe improved sample efficiency and task progression. Qualitative analysis confirms that this
> > agent successfully learns to generate and execute plans at varying levels of abstraction, demonstrating
> > the ability to plan when necessary and replan in response to changing circumstances [Lines 448-451]
> >
> > There is a plethora of analysis that could be done to expand on this statement, such as any kind of analysis on when the model decides to replan, the variance of this frequency, example outputs of replans, such as the model replanning only when something unexpected happens, or if something went wrong. If it is impossible to run the method on a new dataset, then I think that to justify increasing the score to a 6, there should be extensive analysis on the planning behavior of the final model itself in order to increase the manuscript's contribution to the field at large.

---

> > > ### Author Response · Authors · 2025-11-27
> > >
> > > We thank the reviewer for raising their score, and for the helpful and actionable feedback.
> > >
> > > Regarding the typos in Figure 1, we have updated the paper and cleaned up the editing artifacts, thank for spotting this!
> > >
> > > Regarding the condition for raising the score further, we appreciate the reviewer offering two distinct paths. Due to the significant computational resources required to run the full SFT+RL pipeline on a new domain within the short discussion timeframe, we are committing to the reviewer's second option: an extensive analysis of the planning behavior.
> > >
> > > We are currently finalizing a study using a stronger LLM (likely GPT5) as a judge to quantitatively evaluate the SFT vs SFT+RL agent’s planning dynamics. Specifically, we are looking to measure:
> > >
> > > **Plan adherence**: How often the agent successfully carries a plan to completion.
> > > **Reactive replanning**: The frequency of replanning immediately following an unexpected event (e.g., taking damage, failed crafting attempt) versus replanning after a subgoal success.
> > > **Planning statistics**: As requested, we will report the effective planning number (mean and standard deviation of frequency) to quantify the variance.
> > >
> > > We hope to post these results within a few days.

---

> > > > ### Author Response · Authors · 2025-12-03
> > > >
> > > > We thank the reviewer for their continued engagement and for raising their score to a 4. As promised in our previous response, we have now completed the extensive quantitative analysis of the planning behavior to address your condition for raising the score to a 6. Additionally, we have run the fixed-frequency ablation you requested.
> > > >
> > > > ### **Plan every 10 steps ablation**
> > > > To address your question regarding whether a naive prompted model that plans every 10 steps would perform worse, we ran our full RL pipeline with a hard constraint to plan exactly every 10 steps.
> > > >
> > > > Directly answering this question: the model has comparable performance (slightly worse), however, the biggest difference lies in cost. While the fixed agent is forced to plan at a constant rate of 10%, our dynamic model learns to plan much less frequently, approximately 3% of the time, to achieve that same score. We have added this ablation study to the Appendix of the revised paper.
> > > >
> > > > ### **In-Depth Analysis of Planning Behavior**
> > > > To rigorously quantify when and why the agent plans, we collected 100 episodes each of the SFT and SFT+RL agents. We utilized a GPT-5-mini as an LLM judge to evaluate two key metrics: Plan Completion (was the plan successfully carried out?) and Adaptive Replans (was the decision to replan sooner triggered by a justified event, e.g., unexpected failure or danger, rather than random noise?).
> > > >
> > > > | Method | Plan Completion | Adaptive Replans | Avg. Plan Distance | Median Plan Distance |
> > > > |:-------|:---------------:|:----------------:|:------------------:|:--------------------:|
> > > > | SFT    | $0.16 \pm 0.01$ | $0.35 \pm 0.01$  | $13.20 \pm 9.6$    | $12.0 \pm 1.4$       |
> > > > | SFT+RL | $0.20 \pm 0.01$ | $0.41 \pm 0.02$  | $23.29 \pm 23.3$   | $14.0 \pm 9.2$       |
> > > >
> > > > The results in Table above highlight three key behavioral shifts that confirm the agent has learned a meaningful planning policy. First, Plan Completion is significantly higher for the SFT+RL agent ($0.20$ vs $0.16$), indicating that the model has effectively learned to not just generate plans, but to faithfully follow them. Second, the Adaptive Replans ratio is higher ($0.41$ vs $0.35$), suggesting the model has learned to trigger planning adaptively in response to genuine environmental needs rather than random noise.
> > > >
> > > > Third, and perhaps most importantly, the variance in planning intervals is drastically higher for the RL agent (Standard Deviation of $23.3$ vs $9.6$). This high variance confirms that the model plans only when necessary, waiting for long stretches when execution goes smoothly, rather than adhering to the more rigid, fixed-interval behavior observed in the SFT baseline. Collectively, these metrics demonstrate that the dynamic behavior is a learned, state-dependent strategy rather than a random artifact. We have incorporated this table and analysis into the main text of the revised manuscript.
> > > >
> > > > The fixed-frequency ablation confirms that frequent planning can yield good performance; however, our Dynamic agent matches this performance while using approximately 3x less planning compute, validating the efficiency of our approach. Furthermore, the behavioral analysis in Table 2 conclusively demonstrates that this efficiency arises from a non-trivial, adaptive strategy characterized by significantly higher plan completion and justified replanning.
> > > >
> > > > The reviewer kindly mentioned that the above experiments would have been sufficient to justify an increase in their score. However, since we have fulfilled the specific conditions set by the reviewer to increase the score, and the rebuttal period was terminated early due to the ICLR leaks, the reviewer won't be able to further reply or change their score. We kindly ask the Area Chair to consider our further improvements in our response as requested by the reviewer.

---

### Official Review · Reviewer_7gKU · 2025-11-02

**Soundness:** 3
**Presentation:** 3
**Contribution:** 2
**Rating:** 4
**Confidence:** 4

**Summary:**

This paper tackles the important question of when an LLM agent should engage in explicit planning versus acting directly, aiming to improve test-time compute efficiency.
The proposed method introduces a learnable gating policy that decides whether to invoke a planner.
The training procedure consists of two stages:

**1**.	Supervised Fine-Tuning (SFT) on synthetic demonstrations to imitate mixed plan-act behavior;

**2**.	Reinforcement Learning (RL) using a reward that trades off planning benefits against compute costs, regularized by a KL penalty to stabilize learning.

Experiments in synthetic or lightweight environments (POGS, Crafter) show that the learned agent can outperform both “always-plan” and “never-plan” baselines, suggesting a non-trivial tradeoff in reasoning allocation.

**Strengths:**

* This paper raises a relevant and not well explored question about adaptive reasoning allocation in LLM agents.
* This paper provides a conceptually clean formulation that distinguishes between planning and acting decisions.
* The two-stage training (SFT + RL) is straightforward and could serve as a reproducible baseline for future work.
* Preliminary results reveal a potentially meaningful “Goldilocks zone” between under- and over-planning.

**Weaknesses:**

1.	**Limited Scope of Environments**: The paper's broad claims about "LLM Agents" are supported by narrow evidence. The experiments are confined to two non-linguistic, grid-world environments (a synthetic graph search and Crafter), which limits the generalizability of the findings to more complex, language-native agent tasks.
2.	**Insufficient Baseline Comparisons**: The empirical credibility is undermined by weak baselines ("never-plan" and "fixed-frequency"). The paper fails to benchmark against established, relevant reasoning frameworks such as ReAct,ToT, making it difficult to assess the method's true advantage.
3.	**Incomplete Conceptual Framework**: A significant gap exists between the formal framework (Section 3.1) and its practical implementation. Key concepts, like the "Instability Cost" ($C_{noise}$), are described as conceptual but are never measured or explicitly optimized for; they are only assumed to be implicitly penalized.
4.	**Contradictory Ablation Results**: The paper's core premise is challenged by its own cost penalty ablation (Appendix C.3). The results show that while penalties reduce planning frequency, the final task score remains "largely unaffected". This finding questions the practical importance of the very cost-benefit trade-off the paper aims to optimize.
5.	**Missing Sensitivity Analysis**: The RL training relies on key hyperparameters, notably the KL coefficient, to stabilize the policy. The paper lacks a sensitivity analysis for this coefficient, making it unclear how robust the method is or if the policy is prone to degenerating into trivial "always-plan" or "never-plan" behaviors.

**Questions:**

1.	Can the authors formally define and measure the planning advantage metric rather than describing it qualitatively?
2.	How does the method behave under different KL penalty coefficients ($\beta$) — does the policy degenerate into trivial always/never behaviors?
3.	How would the model perform if applied to real reasoning benchmarks such as Mathematical reasoning (e.g., GSM8K, MATH) or program synthesis benchmarks (e.g., HumanEval)?”
4.	Does the gating module generalize across environments, or must it be retrained per task?

---

> ### Author Response · Authors · 2025-11-25
>
> We thank Reviewer 7gKU for their detailed feedback and for highlighting the relevance of our research question, the clean conceptual formulation, and the reproducible nature of our SFT+RL baseline. We are glad you found the "Goldilocks zone" finding to be meaningful. We address your specific concerns and questions below.
>
> ### **Clarification on planning policy**
> First, regarding your question about the generalizability of the "gating policy," we would like to clarify that our method does not introduce a separate, specialized gating module or policy that requires distinct training. Instead, the decision to plan is learned implicitly by the single, monolithic LLM agent itself. As described in Section 3, our agent realizes its conceptual policies through a unified output format; it implicitly decides to plan simply by generating the <plan> token as the next token in its sequence. This unified approach is a core part of our design, allowing the agent to flexibly learn when to allocate compute without architectural changes, meaning nothing prevents this mechanism from being applied in general multi-task settings. We have revised the text to make this distinction more prominent in section 3.
>
> ### **Single-step vs long-horizon sequential tasks**
> We also acknowledge your concern regarding the scope of our environments. We intentionally selected POGS and Crafter because they represent challenging, long-horizon (T>>1), partially-observable environments where the decision of "when to plan" is both non-trivial and impactful. Our method is specifically designed for such sequential tasks where planning at every step is computationally wasteful and can degrade performance, a phenomenon we empirically demonstrate as the "Goldilocks" effect. In contrast, for single-step reasoning benchmarks like GSM8K or MATH, the interaction is typically a single pass (T=1). In such contexts, there is no long-horizon budget to manage, and allocating full compute (i.e., "always plan" or Chain-of-Thought) is typically optimal. Therefore, our framework is best evaluated on sequential decision-making problems where an agent must dynamically budget its compute over time. We have clarified this scope and the distinction from single-step reasoning tasks in the paper.
>
> ### **ReAct and ToT**
> Regarding baselines, we appreciate the suggestion to compare against established frameworks like ReAct and Tree of Thoughts (ToT). We would like to clarify that our "always-plan" baseline is identical to the ReAct framework, as it prompts the model to explicitly reason (think) before every single action. Regarding ToT, we view it as a powerful method for how to plan during a single reasoning step (exploring multiple paths), whereas our work focuses on when to plan (deciding whether to allocate compute for a plan at all). We view ToT as orthogonal to our core research question of dynamic temporal compute allocation, and we have added a sentence to the related work to clarify this distinction.

---

> > ### Author Response · Authors · 2025-11-25
> >
> > ### **Planning advantage and instability cost**
> > You also raised an important question regarding the formalization of the "Planning Advantage" and "Instability Cost." We clarify that the Planning Advantage is a conceptual tool, analogous to the standard advantage function in RL, representing the theoretical, state-dependent expected benefit of planning. It is not a value we measure directly; rather, it is the value that our RL agent implicitly learns to approximate when it optimizes its policy to maximize reward. Similarly, the "Instability Cost" ($C_{noise}$​) is a conceptual term we use to formalize the performance degradation observed from excessive planning (e.g., oscillation or backtracking in POGS). The agent learns to avoid this behavior because it naturally leads to lower task rewards, not because we feed it an explicit instability penalty. We have revised Section 3 to make it clearer that these are theoretical motivations for the behavior optimized by the RL objective.
> >
> > ### **Penalty cost ablation**
> > Regarding the cost penalty ablation in Appendix C.3, we thank the reviewer for this sharp observation. The results are indeed subtle, and we have clarified their implication. The reviewer is correct that after RL training, penalizing planning tokens (even heavily) does not significantly degrade the final autonomous score. This is because, as we hypothesise in the appendix, the agent has internalized the effective policies. However, the benefit of training the agent to plan explicitly is not just its autonomous score, but the steerability and collaboration this capability unlocks. As shown in Figures 3, 11, and 15, the agent trained to generate plans could be steered by human-provided plans to achieve the game-winning task of collecting diamonds, a feat no autonomous agent (planning or not) could accomplish in our experiments. Therefore, the cost-benefit trade-off is not just about autonomous score vs. tokens. It's about training an agent that is both efficient (learns to internalize plans and not over-plan) and steerable (can follow external plans), which is a primary contribution of our work. We have moved this discussion to the main paper to better articulate this trade-off.
> >
> > ### **Sensitivity to beta of KL divergence**
> >
> > Finally, to address your concern about hyperparameter sensitivity, we performed a sweep of the KL coefficient (beta [0.05, 0.1, 0.15, 0.2, 0.3]) trained over 200k environment steps without learning rate warmup to stress-test stability. The results (added to Appendix C) demonstrate that our method is robust to hyperparameter choice within the [0.1, 0.2] range. We identified beta=0.1 as a 'sweet spot'. We observed two failure modes at the extremes: setting beta too low (0.05) causes the planning behavior to collapse, degenerating into a policy that plans only plans once, usually at the first time step, in a sort of knowledge dump, while setting it too high (0.3) constrains the policy too heavily, where we observe that the model is still making a lot of invalid actions (invalid action frequency).
> >
> > Given these clarifications regarding baselines and formal definitions, and the sensitivity analysis, we hope you might reconsider your score.

---

### Official Review · Reviewer_jvQS · 2025-11-02

**Soundness:** 3
**Presentation:** 3
**Contribution:** 3
**Rating:** 6
**Confidence:** 3

**Summary:**

The paper studies how an agent can dynamically decide when to spend test-time compute on planning. The core idea is to treat planning as a state-dependent cost–benefit trade-off: plan when the expected value gain (a “planning advantage”) exceeds the costs (tokens/latency/instability). The training recipe has two stages: SFT priming on trajectories where a teacher plans at varied intervals, so the student learns both the format and the option to plan or not; then RL fine-tuning (PPO) to optimize task reward, optionally with a planning-cost penalty, so the agent learns when planning is worthwhile. In Crafter (and a synthetic POGS environment), the SFT+RL dynamic planners are more sample-efficient than non-planning baselines and can be steered by human-written plans to reach achievements beyond their solo ability.

**Strengths:**

1. The work isolates the when-to-plan decision, reports a “Goldilocks” effect (best performance at intermediate planning frequency), and motivates learning-based control of test-time compute rather than fixed schedules.

2. After RL, planning agents are more sample-efficient and reach more complex achievements than matched non-planning baselines; they can also be human-steered post-RL.

3. Zero-shot analyses with Llama-3.3-70B-Instruct in POGS/Crafter support the non-monotonic compute–performance relationship

**Weaknesses:**

1. The RL experiments fine-tune Llama-3.1-8B-Instruct; broader open-source families (e.g., Qwen/Mistral/OLMo) under the same RL protocol would better test robustness.

2. RL results are centered on Crafter. Additional partially observable, long-horizon tasks (e.g., ALFWorld) would test generality.

3. What is the training cost?

**Questions:**

See Weaknesses.

---

> ### Author Response · Authors · 2025-11-25
>
> We thank Reviewer jvQS for their positive and accurate summary of our work. We are glad the reviewer found our contributions valuable, particularly the isolation of the "when-to-plan" decision, the identification of the "Goldilocks" effect, and the demonstration that our SFT+RL agents are more sample-efficient and human-steerable.
>
> ### **On the scope and generality concerns**
> The reviewer's main concerns focus on the scope of our experiments (model families and environments) and the associated training cost. These points are closely related, and we will address them together. We agree with the reviewer that our experimental scope, which is focused on Llama-3.1-8B-Instruct and the POGS/Crafter environments, is a key limitation. To address this, we conducted additional zero-shot experiments on an entirely new environment TextWorld, (using the three environments taken from Balrog, similar in style to ALFworld), as well as with a new model family (Gemini-2.5-Flash). Results can be seen in appendix C.5 in the paper. In both cases, we confirmed that the core "Goldilocks" finding holds true, suggesting our findings are robust beyond the specific setups in the main paper.
>
> ### **Training cost and limitations**
> The reviewer's third question, "What is the training cost?", is the precise reason for this limited scope. A single SFT+RL experiment run, as described in Appendix C, typically completed within 24-48 hours using 8xH100 GPUs (6 GPUs for training, 2 for data collection). However, as the reviewer is likely aware, the true computational cost of this research is not in a single run, but in the extensive experimentation and hyperparameter search required to find a stable and effective agent. This process would have to be repeated for any new model or environment. This substantial computational investment is exactly why our current study was constrained in scope. We believe our work establishes a strong baseline, and we hope our findings (and released code) will enable future work to explore the robustness and generality that the reviewer rightly points to.
>
>
> Given these clarifications and our new results on TextWorld, we kindly ask if you would consider increasing your score.

---

### Author Response · Authors · 2025-11-25
**General response**

We thank all Reviewers for their thoughtful and constructive comments on our work. We are encouraged that the reviewers agree our paper addresses a relevant, important, and critical problem in existing LLM agents: learning when to allocate test-time compute for planning. Reviewers appreciated our conceptual cost-benefit framework, the reproducible SFT+RL pipeline, and our "Goldilocks" finding that motivates a learning-based approach over fixed strategies. Based on the feedback, we have identified three main areas that require additional clarification and improvement. We address these through new experiments and analyses conducted during the rebuttal period.

### **Relation to Prior Work**

A concern from some Reviewers was our claim of this work being the first to explore the problem of when to allocate test-time compute in sequential decision making tasks, in context of prior work like Hu et al. (2024) and UI-Tars (2025). We acknowledge that our initial language may have been too absolute, and we have updated the abstract and related work section to clarify distinctions between these valuable contributions. We emphasize however a fundamental distinction: Hu et al. (2024) employ a separate 'Mediator' policy to trigger an external planner (effectively tool use), and UI-Tars utilizes SFT to merely imitate a dynamic behavior. In contrast, our work is the first to fine-tune a single LLM to implicitly learn a state-dependent cost-benefit trade-off using RL optimization, deciding when the expected value gain from planning justifies the computational cost. This unified, RL-optimized approach represents a novel and foundational step for autonomous LLM agents. We have updated our abstract and related work section to explicitly clarify these distinctions.


### **Scope of Experiments, Baselines, and Robustness**

Reviewers noted that our fine-tuning experiments were focused on POGS and Crafter. While computational constraints (detailed in our response to R1) prevent running the full SFT+RL pipeline on entirely new domains, we have addressed the concern regarding generalizability by conducting additional zero-shot experiments on TextWorld using Llama-3.3-70B and Gemini-2.5-Flash. Results can be seen in appendix C.5 of the paper. We observed the same non-monotonic "Goldilocks" relationship between planning frequency and performance reported in the main paper, confirming the phenomenon is not an artifact of our specific setup. Furthermore, to address requests for direct comparisons, we have added a summary table (Table 1) to the main paper that explicitly compares Zero-shot, SFT, and SFT+RL performance and token costs side-by-side, demonstrating that our 8B model outperforms the 70B zero-shot baseline while being significantly more token-efficient. Finally, regarding hyperparameter sensitivity (R2), we completed a sensitivity analysis on the KL coefficient (beta [0.05, 0.1, 0.15, 0.2, 0.3]) over 200k environment steps. We found the method to be robust, with consistent task progression across intermediate values. We identified beta=0.1 as the optimal balance, effectively preventing policy degeneration (where planning disappears at beta=0.05) while avoiding over-regularization (where invalid actions persist at beta=0.3). We have added these results to Appendix C.


### **Clarification of Framework and Ablation Results**
Some reviewers raised questions regarding the "Instability Cost" and "Plan Drift." We clarify that these are theoretical motivations for why dynamic planning outperforms static schedules, rather than explicit terms in the loss function. The RL agent implicitly learns to avoid these costs by optimizing the task reward. Regarding the cost penalty ablation, Reviewers noted that higher penalties reduced planning frequency without significantly degrading scores. We clarify that this indicates the agent successfully internalizes the planning policy, learning to act optimally without needing to output explicit planning tokens. However, we emphasize that the explicit planning capability remains crucial for the steerability and human-AI collaboration results shown in our qualitative analysis, which would be lost in a purely internalized "no-plan" policy. We have moved this discussion to the main paper to better articulate this trade-off. We thank the Reviewers again for their valuable feedback.


### **Citations**
Bin Hu, Chenyang Zhao, Pu Zhang, Zihao Zhou, Yuanhang Yang, Zenglin Xu, and Bin Liu. Enabling intelligent interactions between an agent and an llm: A reinforcement learning approach, 2024. URL https://arxiv.org/abs/2306.03604.

Yujia Qin, Yining Ye, Junjie Fang, Haoming Wang, Shihao Liang, Shizuo Tian, Junda Zhang,
[...] Xin Liu, and Guang Shi. Ui-tars: Pioneering automated gui interaction
with native agents, 2025. URL https://arxiv.org/abs/2501.12326.

---

> ### Author Response · Authors · 2025-12-03
> **Final general response**
>
> Dear Area Chair,
>
> Due to the recent OpenReview leaks, which terminated the discussion period early and reset scores, we are submitting this summary to discuss progress made just before the interruption. While our previous response addressed general concerns regarding prior work and experimental scope, we wish to present significant new experimental results that quantitatively validate our method's efficacy.
>
> These analyses focus on the agent's planning behavior,  which is a key aspect that reviewer **dRJY** asked as final requisite for raising their score to a 6 (having already moved from 2 to 4). We have now successfully completed this analysis, filling a critical gap identified during the discussion.
>
> ### **New Planning Analysis**
>
> We have now completed this analysis and added it in section 5 and appendix C6 of the paper, improving the quality of our work by confirming that the agent’s dynamic behavior is a learned, adaptive strategy. Specifically, our new results demonstrate that the RL-trained agent achieves significantly higher plan completion (0.20 vs 0.16) and a higher rate of adaptive replanning (0.41 vs 0.35) compared to the SFT-only fine-tuned model. Furthermore, we conducted the requested "fixed-frequency" ablation, which confirmed that while a naive agent forced to plan every 10 steps has almost comparable performance, our dynamic agent achieves this same score while utilizing approximately 3x less planning compute.
>
> Since we have fulfilled the specific experimental conditions set by Reviewer dRJY for a score of 6, we respectfully ask the Area Chair to consider this intended adjustment in their final decision. Although the openreview issues prevented the other reviewers from responding on time, we believe that the strength of our rebuttal, combined with these new findings regarding efficiency and adaptive behavior, would likely have led to similar score adjustments across the board. We hope you will weigh these new empirical findings and the positive trajectory of the discussion that was unfortunately cut short.
>
> Kind regards,
>
> The authors.

---

### Meta-Review · Area_Chair_pPz2 · 2026-01-06

**Summary:**

The paper proposes a framework for training LLM agents to dynamically allocate test-time compute for planning via a two-stage SFT and RL pipeline. While reviewers found the problem formulation timely and appreciated the "Goldilocks" insight regarding optimal planning frequency, the decision to reject is driven by the limited experimental scope. The empirical validation relies heavily on the Crafter environment and smaller models, which reviewers found insufficient to support the paper's broad claims about generalizable agent capabilities. Additionally, concerns regarding the strength of baselines and the distinction from prior work on dynamic planning suggest the paper requires more rigorous evaluation on diverse benchmarks to be ready for publication.

**Reviewer Concerns:**

The rebuttal successfully addressed several technical requests, including a sensitivity analysis for the KL penalty and a detailed comparison of token efficiency. The authors also provided a late-breaking analysis of planning behavior (completion rates and adaptive replanning) that directly responded to specific feedback from Reviewer dRJY. However, the most critical concern regarding the breadth of evaluation remains outstanding. Multiple reviewers emphasized the need for validation beyond the Crafter grid-world; while the authors provided zero-shot evidence on TextWorld, the lack of full training experiments on distinct, complex domains means the robustness of the proposed learning method remains unproven.

**Reviewer Scores:**

Reviewer dRJY engaged actively and indicated a willingness to raise their score conditional on additional behavioral analysis, which the authors provided in their final comment. Reviewers 7gKU and Fq61 did not respond to the rebuttal. Given that their primary concerns regarding the limited environmental scope were not fully addressed by the new zero-shot experiments, their scores would likely have remained unchanged.

---

### Decision · Program_Chairs · 2026-01-26

Reject